# Feedback Control for Multi-Objective Graph Self-Supervision

**Karish Grover** [1 2]  **Theodore Vasiloudis** [3]  **Han Xie** [3]  **Sixing Lu** [3]  **Xiang Song** [3]  **Christos Faloutsos** [1]

## Abstract

Can multi-task self-supervised learning on graphs be coordinated without the usual tug-of-war between objectives? Graph self-supervised learning (SSL) offers a growing toolbox of pretext objectives—mutual information, reconstruction, contrastive learning—yet combining them reliably remains a challenge due to objective interference and training instability. Most multi-pretext pipelines use *per-update mixing*, forcing every parameter update to be a compromise, leading to three failure modes: *Disagreement* (conflict-induced negative transfer), *Drift* (nonstationary objective utility), and *Drought* (hidden starvation of underserved objectives). We argue that coordination is fundamentally a *temporal allocation* problem: deciding *when* each objective receives optimization budget, not merely how to weigh them. We introduce `ControlG`, a control-theoretic framework that recasts multi-objective graph SSL as *feedback-controlled temporal allocation* by estimating per-objective difficulty and pairwise antagonism, planning target budgets via a Pareto-aware log-hypervolume planner, and scheduling with a Proportional–Integral–Derivative (PID) controller. Across 9 datasets, `ControlG` consistently outperforms state-of-the-art baselines, while producing an auditable schedule that reveals *which objectives drove learning*.

## 1. Introduction

Self-supervised pretraining has become a default route to transferable representations when labels are scarce, and graphs are no exception. Graph SSL has produced a diverse toolbox of objectives encoding different inductive biases: mutual-information maximization (Veličković et al., 2018), contrastive learning under augmentations (You et al., 2020;

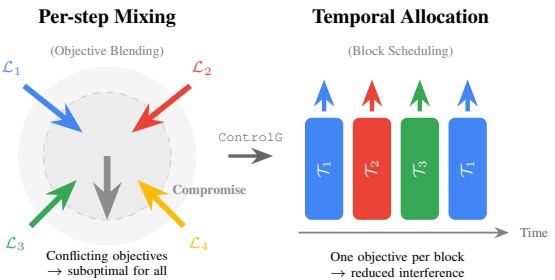

*(a)* **The core insight.** Per-step gradient mixing forces a compromise when objectives conflict. `ControlG` separates objectives across time, eliminating per-step interference by design.

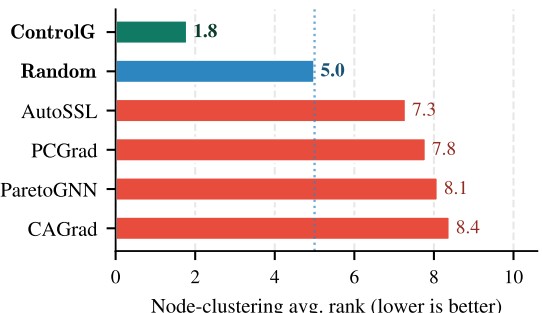

*(b)* **Temporal separation is a strong inductive bias.** On node clustering, a *uniformly random* single-task scheduler (blue) already outranks every per-step mixing method (red), and `ControlG` (teal) widens the gap further still.

*Figure 1.* **Motivation for** `ControlG`. **(a)** Per-step gradient mixing forces an instantaneous compromise among conflicting objectives; temporal allocation gives each its own blocks. **(b)** Empirically, even unplanned temporal separation beats sophisticated per-step mixing, motivating feedback-controlled scheduling.

Zhu et al., 2020), negative-free bootstrapping (Thakoor et al., 2021), and masked reconstruction (Hou et al., 2022). Yet *which* objective works best depends heavily on the dataset, downstream task, and training stage—motivating multi-pretext approaches that combine or search over objectives (Jin et al., 2021).

A natural response is to train with multiple pretext objectives jointly, hoping that the encoder absorbs a richer set of signals. Many multi-pretext and multi-task recipes operationalize this by optimizing a weighted sum of losses or by combining gradients into a single update direction

---

[1]Carnegie Mellon University [2]The work was done during Karish Grover's internship at Amazon, US. [3]Amazon. Correspondence to: Karish Grover <karishg@cs.cmu.edu>.

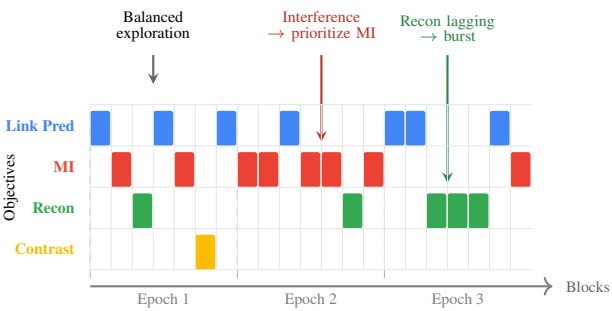

*Figure 2.* **Interpretable schedule.** `ControlG` produces an auditable allocation pattern: early training explores all objectives; mid-training prioritizes MI after detecting interference; late-training bursts on lagging reconstruction. The planner adapts allocation based on interference and demand signals.

at every step. In graph SSL specifically, ParetoGNN (Ju et al., 2022) uses multi-gradient descent (MGDA) (Sener & Koltun, 2018) to reconcile multiple pretext tasks, and WAS (Fan et al., 2024) proposes an instance-level framework that decouples selecting and weighing tasks. In general multi-task learning, a large body of work designs per-step scalarization rules using MGDA or mitigates interference through gradient manipulation (Yu et al., 2020; Liu et al., 2021). These approaches are effective in many settings, but they still share a core actuation pattern: each training step blends signals across objectives in parameter space, implicitly assuming that an appropriate instantaneous compromise can be found and maintained.

**Research gaps in multi-objective graph SSL.** When objectives are self-supervised pretext tasks, training exhibits three recurring failure modes. **(a)** *Drift*: the utility of a pretext objective is nonstationary—an objective that accelerates learning early can become redundant later—so fixed or weakly adaptive weights lag behind regime changes (Guo et al., 2018). **(b)** *Disagreement*: objectives frequently induce conflicting gradient directions, forcing a compromise that can be simultaneously suboptimal for all (Sener & Koltun, 2018; Yu et al., 2020; Liu et al., 2021). **(c)** *Drought*: adaptive weighting can starve objectives by driving their weights toward zero, making it unclear whether an objective ever meaningfully shaped the representation. Together, these issues make multi-pretext training brittle: current systems are largely *open-loop*, specifying a mixing rule and hoping the dynamics remain well-behaved. Strikingly, the instantaneous compromise of per-step mixing is so harmful that even *unplanned* temporal separation—a uniformly random single-task scheduler—outranks sophisticated per-step MTL methods on downstream clustering (Fig. 1b), motivating our feedback-controlled approach.

> **Key idea.** Instead of blending all self-supervised objectives at every step, train *one objective at a time* and let feedback control decide which objective is active in each block.

`ControlG` addresses these gaps by coordinating objectives through *temporal allocation* under a fixed compute budget. At a slower sensing timescale, it performs *state estimation on the full graph* to obtain two signals per objective: (i) a *spectral demand* indicator that measures how rapidly the objective's learning signal varies over the graph and is motivated by the spectral/low-pass view of message passing (Nt & Maehara, 2019) (e.g., simplified GCN propagation corresponds to a fixed low-pass filter); and (ii) an *interference* indicator derived from local multi-objective gradient geometry, where MGDA provides Pareto-relevant weights for which objectives are actively constraining the trade-off at the current iterate. Based on these state estimates, a *Pareto-aware planner* produces a target allocation using log-hypervolume sensitivities, leveraging the hypervolume indicator's standard role as a Pareto-compliant scalarization (Guerreiro et al., 2020). Finally, a *tracking controller* converts the continuous allocation target into discrete task choices by tracking allocation deficits with a PID-style feedback law; this deficit perspective is closely related to classic deficit-counter scheduling ideas (e.g., deficit round robin (Shreedhar & Varghese, 1996)), which ensure long-run fairness when service is indivisible. The end result is an interpretable, auditable training process: the planner explains *why* an objective should receive budget (priority + state), and the controller ensures it actually receives that budget over time. We hereby summarize our contributions:

- We introduce `ControlG`, a closed-loop control framework for coordinating multiple graph SSL pretext objectives via sensing, Pareto-aware planning, and feedback-based execution. To the best of our knowledge, this is the first work to cast multi-objective graph SSL coordination as a *closed-loop scheduling problem with explicit allocation tracking*, using deficit-based feedback control to execute discrete single-objective blocks rather than per-step loss weighting.

- We propose full-graph state estimation signals that quantify objective demand and objective interference, enabling principled adaptation to nonstationarity (drift) and gradient conflicts (disagreement) without collapsing all objectives into a single per-step compromise.

- We develop a Pareto-aware planner based on log-hypervolume sensitivities and a deficit-tracking PID controller that executes discrete single-task blocks while explicitly mitigating starvation (drought).

- We empirically evaluate `ControlG` against strong single-objective and multi-pretext baselines on stan-

dard graph SSL benchmarks (Sec. 5), showing improved robustness and transfer.

**Conflict of Interest Disclosure.** This work was carried out as academic research. Several authors are affiliated with Amazon, and the first author completed part of this work during an internship at Amazon; the remaining work was conducted at Carnegie Mellon University. `ControlG` is a research method introduced in this paper and is not a commercial product or deployed system; all baselines are publicly available academic methods, and no proprietary or commercially deployed system is evaluated. The authors declare no other financial conflicts of interest.

## 2. Related Work

**Graph SSL and multi-pretext integration.** Graph SSL has produced diverse pretext objectives: contrastive methods like DGI (Veličković et al., 2018), GraphCL (You et al., 2020), and GCA (Zhu et al., 2021); bootstrapping approaches like BGRL (Thakoor et al., 2021); and reconstruction objectives like GraphMAE (Hou et al., 2022). Since no single objective dominates across graphs (Liu et al., 2022; Ju et al., 2024), recent work integrates multiple objectives. AUTOSSL (Jin et al., 2021) searches task combinations via label-free surrogates but uses static weights once selected, limiting adaptivity to training dynamics. PARETOGNN (Ju et al., 2022) applies multi-objective gradient updates for Pareto efficiency, yet per-step mixing can force compromise when objectives conflict and provides no explicit coverage accounting. WAS (Fan et al., 2024) decouples selection and weighting at the instance level but still blends all objectives per update. GraphTCM (Fang et al., 2024) models task correlations, confirming they vary across datasets—motivating our adaptive, feedback-driven approach. Unlike these methods, `ControlG` coordinates via *temporal allocation*: time-sharing single-task blocks with explicit deficit tracking, avoiding forced compromise while keeping coverage of every objective auditable across training.

**Multi-objective optimization and MTL.** Gradient-based MTL methods design per-step update rules: MGDA finds minimum-norm gradient combinations (Sener & Koltun, 2018), PCGrad/CAGrad reduce interference (Yu et al., 2020; Liu et al., 2021), and Nash-MTL uses bargaining (Navon et al., 2022). Imbalance-focused approaches include GradNorm (Chen et al., 2018) and uncertainty weighting (Kendall et al., 2018). These methods address conflict through instantaneous scalarization but still produce a single update direction, making it difficult to separate objectives in time or provide block-level coverage accountability. `ControlG` instead uses gradient geometry for *sensing* (interference feedback) while actuating through time-sharing.

**Task scheduling and curricula.** Dynamic Task Prioritization (Guo et al., 2018) and AdaTask (Laforgue et al., 2022) adapt *when* to train tasks, but are not developed for graph SSL and lack Pareto-aware planning. Without explicit allocation tracking, schedules can still exhibit hidden starvation. `ControlG` combines a Pareto-aware planner with deficit-tracking control, yielding interpretable time-sharing with explicit coverage tracking and a probabilistic drought bound.

## 3. Background and Preliminaries

We fix notation and recall the primitives `ControlG` builds on: multi-objective gradient geometry, spectral smoothness, hypervolume-based scalarization, and PID control.

**Setup and notation.** Let $G = (V, E, \mathbf{X})$ be an attributed graph with $n = |V|$ nodes, adjacency $\mathbf{A} \in \{0,1\}^{n \times n}$, degree matrix $\mathbf{D} = \mathrm{diag}(d_1, \ldots, d_n)$, and node features $\mathbf{X} \in \mathbb{R}^{n \times d}$. A graph encoder $f_\theta$ produces embeddings $\mathbf{Z} = f_\theta(G) \in \mathbb{R}^{n \times h}$. We consider $K$ pretext tasks $\{\mathcal{T}_k\}_{k=1}^K$ with losses $\{\mathcal{L}_k(\theta)\}_{k=1}^K$, treating training as minimizing the vector objective $\boldsymbol{\mathcal{L}}(\theta) = (\mathcal{L}_1, \ldots, \mathcal{L}_K) \in \mathbb{R}^K$.

**Pareto optimality and MGDA.** A point $\theta$ *Pareto-dominates* $\theta'$ if $\mathcal{L}_k(\theta) \leq \mathcal{L}_k(\theta')$ for all $k$ with strict inequality for at least one $k$; we call $\theta$ *Pareto-optimal* if no other point dominates it. *Pareto stationarity*—a first-order necessary condition—holds when $\exists \lambda \in \Delta^K$ s.t. $\sum_k \lambda_k \nabla_\theta \mathcal{L}_k = \mathbf{0}$, where $\Delta^K$ is the probability simplex. Intuitively, no descent direction can simultaneously decrease every objective at this point.

The multi-gradient descent algorithm (MGDA) finds the minimum-norm convex combination of gradients (Sener & Koltun, 2018):

$$\lambda^\star \in \arg \min_{\lambda \in \Delta^K} \left\| \sum_{k=1}^K \lambda_k g_k \right\|_2^2, \quad g_k := \nabla_\theta \mathcal{L}_k. \quad (1)$$

When $\sum_k \lambda_k^\star g_k \neq \mathbf{0}$, this yields a common descent direction that improves all objectives. We use MGDA weights $\lambda^\star$ as an interference signal: objectives with high weight are actively constraining the trade-off at the current iterate.

**Spectral smoothness.** We use the normalized Laplacian $\tilde{\mathbf{L}} := \mathbf{I} - \mathbf{D}^{-1/2} \mathbf{A} \mathbf{D}^{-1/2}$ to quantify signal smoothness on the graph. For a signal $\mathbf{H} \in \mathbb{R}^{n \times d}$, the Dirichlet energy $\mathcal{E}(\mathbf{H}) = \mathrm{tr}(\mathbf{H}^\top \tilde{\mathbf{L}} \mathbf{H})$ measures variation across edges. Since this scales with $\|\mathbf{H}\|_F^2$, we use the Rayleigh quotient $\mathrm{RQ}(\mathbf{H}) := \mathcal{E}(\mathbf{H}) / \|\mathbf{H}\|_F^2$, interpretable as average graph frequency (Von Luxburg, 2007). Because GNNs act as low-pass filters (Nt & Maehara, 2019), objectives inducing high-RQ signals are spectrally harder to optimize.

**Dominated hypervolume.** To scalarize multi-objective progress while remaining sensitive to trade-offs, we use the log-hypervolume indicator (Guerreiro et al., 2020). For a reference point $\mathbf{r} \in \mathbb{R}^K$ with $r_k > \mathcal{L}_k$ (worse than current values), $\log \mathcal{H} = \sum_k \log(r_k - \mathcal{L}_k)$ decomposes additively and is numerically stable. The gradient $\partial \log \mathcal{H} / \partial \mathcal{L}_k = -1/(r_k - \mathcal{L}_k)$ shows that proximity to the reference amplifies sensitivity, so the planner naturally allocates more budget to objectives that lag furthest behind.

**PID control and deficit tracking.** We track target allocations via discrete-time PID control. Let $e_k(m) = N_k^{\text{ref}}(m) - N_k(m-1)$ be the *pre-decision* deficit (target minus actual count *before* block $m$ is assigned) for task $k$, where $N_k^{\text{ref}}(m) = m f_k$. The controller output is:

$$\nu_k(m) = K_P e_k(m) + K_I \sum_{\tau \leq m} e_k(\tau) + K_D \Delta e_k(m),$$

where $\Delta e_k(m) = e_k(m) - e_k(m-1)$. We map $\boldsymbol{\nu}(m)$ to probabilities via softmax. This deficit-tracking view parallels deficit round robin scheduling (Shreedhar & Varghese, 1996), which accumulates "service credit" to ensure long-run fairness when service is discrete.

# 4. ControlG: Overview

The core idea behind ControlG is to replace per-step objective blending with *temporal allocation*: rather than forcing every parameter update to compromise across all objectives, we dedicate consecutive optimization steps to a single objective at a time and use feedback control to decide which objective to train next. This reframes multi-objective coordination as a closed-loop scheduling problem, where the learning system repeatedly (i) *senses* which objectives are currently difficult or mutually antagonistic for the shared encoder, (ii) *plans* an explicit target allocation over objectives using a Pareto-monotone progress surrogate, and (iii) *tracks* that plan with a lightweight controller. Figure 3 illustrates this three-loop architecture. The time-sharing view exposes allocation and ordering as first-class degrees of freedom, yields interpretable schedules, and keeps the inner-loop optimizer unchanged.

Formally, let $\{L_k(\theta)\}_{k=1}^K$ be SSL objectives sharing encoder parameters $\theta$. Training is organized into epochs $t \in \{1, \ldots, T\}$, each containing $B_{\text{ep}}$ mini-batches. We group consecutive mini-batches into *blocks* of size $B_{\text{block}}$, yielding $M := \lceil B_{\text{ep}} / B_{\text{block}} \rceil$ blocks per epoch, indexed by $m \in \{1, \ldots, M\}$. At block $(t, m)$ the scheduler selects a task index $k_{t,m} \in \{1, \ldots, K\}$ and performs $B_{\text{block}}$ standard optimizer steps on that task only,

$$\text{for } j = 1, \ldots, B_{\text{block}}: \quad \theta \leftarrow \theta - \eta \nabla_\theta L_{k_{t,m}}(\theta; \xi_{t,m,j}), \tag{2}$$

where $\xi_{t,m,j}$ denotes the randomness in sampling and augmentations for the $j$-th mini-batch within block $m$. The key

question is therefore not how to combine objectives within a step, but how to select the sequence $\{k_{t,m}\}$ so that progress across pretext objectives stays balanced over time. We answer this by maintaining a compact per-task state computed on the full training graph (Sec. 4.1), converting that state into an allocation target $f(t) \in \Delta^K$ (Sec. 4.2), and tracking $f(t)$ through a deficit-based controller (Sec. 4.3).

## 4.1. State Estimation on the Full Graph

ControlG maintains a compact per-objective state that is updated every $u$ blocks and consumed by the planner (Sec. 4.2) and controller (Sec. 4.3). State estimation is performed on the *full training graph* $G = (V, E)$ so that the signals correspond to the same global objectives being optimized. At each sensing step, for every objective $k \in \{1, \ldots, K\}$ we estimate (i) a *spectral demand* score that captures how rapidly the objective's learning signal varies over the graph, (ii) an *interference* score that captures conflicts with other objectives in parameter space, and (iii) a normalized loss $\tilde{L}_k$ used for planning. We combine (i)–(ii) into a single bounded difficulty state $D_k$ that tempers allocation decisions. All derivations and formal guarantees are provided in App. A.

### 4.1.1. SPECTRAL DEMAND FROM REPRESENTATION-GRADIENT VARIATION

To define a graph-frequency notion we require a node-indexed signal. Parameter gradients are not naturally node-indexed, so we use the objective's gradient with respect to node embeddings. Let $z_v \in \mathbb{R}^h$ denote the embedding of node $v \in V$, and for objective $k$ define the *representation-gradient field* $h_k(v) := \nabla_{z_v} L_k \in \mathbb{R}^h$, stacking rows into $H_k \in \mathbb{R}^{n \times h}$. Let $\tilde{\mathbf{L}}$ be the symmetric normalized Laplacian.[1] The *Dirichlet energy* $\mathcal{E}_k := \text{tr}(H_k^\top \tilde{\mathbf{L}} H_k)$ measures edge-wise variation (App. A.2). Since $\mathcal{E}_k$ scales with gradient magnitude, our spectral-demand state is the scale-invariant *Rayleigh quotient*

$$\text{RQ}_k(\theta) := \frac{\mathcal{E}_k(\theta)}{\|H_k\|_F^2 + \varepsilon}, \tag{3}$$

which, as shown in App. A.3, equals an energy-weighted average Laplacian eigenvalue of $H_k$; larger $\text{RQ}_k$ indicates that the learning signal concentrates on higher graph frequencies (sharper variation across edges). Why interpret $\text{RQ}_k$ as a *difficulty proxy*? Many message-passing GNNs behave as low-pass graph filters (Nt & Maehara, 2019), attenuating higher-frequency components, which directly limits attainable progress:

**Proposition 4.1** (Spectral demand bounds attainable progress; informal, see Prop. A.8 in App. A.4)**.** *Assume*

---

[1] $\tilde{\mathbf{L}} = \mathbf{I} - \mathbf{D}^{-1/2} \mathbf{A} \mathbf{D}^{-1/2}$; for directed graphs we work with the symmetrized adjacency $(\mathbf{A} + \mathbf{A}^\top)/2$.

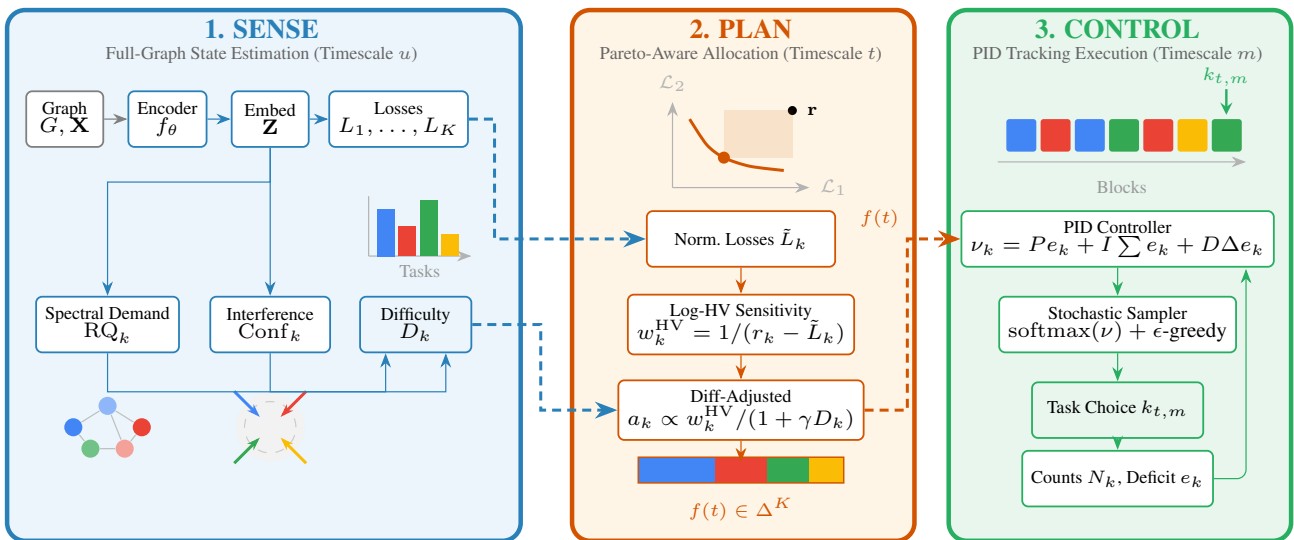

*Figure 3.* **Overview of** `ControlG`. The framework decomposes multi-task graph self-supervised learning into three coupled loops operating at different timescales. **(1) SENSE** (timescale $u$): A shared encoder $f_\theta$ maps the graph $(G, \mathbf{X})$ to embeddings $\mathbf{Z}$, from which per-task losses $L_k$ are computed. State estimation derives two difficulty signals—*Spectral Demand* ($\mathrm{RQ}_k$), measuring how high-frequency the learning signal is on the graph, and *Interference* ($\mathrm{Conf}_k$), measuring gradient conflicts with other Pareto-relevant objectives—which are combined into a composite difficulty state $D_k$. **(2) PLAN** (timescale $t$): Normalized losses $\tilde{L}_k$ are converted to *Log-HV sensitivities* $w_k^{\mathrm{HV}} = 1/(r_k - \tilde{L}_k)$, quantifying each objective's marginal contribution to Pareto progress. These are tempered by difficulty to produce a *difficulty-adjusted allocation* $a_k \propto w_k^{\mathrm{HV}}/(1 + \gamma D_k)$, normalized to an epoch-level target $f(t) \in \Delta^K$. **(3) CONTROL** (timescale $m$): A PID controller tracks the allocation plan by computing logits $\nu_k$ from deficits $e_k = N_k^{\mathrm{ref}} - N_k$, their integral, and derivative. A stochastic sampler (softmax with $\epsilon$-greedy exploration) selects the task $k_{t,m}$ for each block, updating counts $N_k$ and closing the feedback loop. Dashed arrows indicate cross-module information flow.

*representation updates act as a low-pass graph filter. Then, for fixed gradient energy $\|H_k\|_F^2$, the first-order loss decrease per update is upper-bounded by a quantity that is non-increasing in* $\mathrm{RQ}_k$.

#### 4.1.2. INTERFERENCE VIA PARETO-RELEVANT GRADIENT GEOMETRY

Spectral demand captures graph-structural difficulty but not conflicts from shared parameters. We therefore measure interference in parameter space. Let $g_k := \nabla_\theta L_k$ be the parameter gradient for objective $k$. A first-order expansion shows that a step on $k$ increases $L_j$ whenever $\langle g_j, g_k \rangle < 0$ (Claim A.10 in App. A.5), motivating negative cosine as a conflict indicator. To focus on Pareto-relevant objectives, we use MGDA as a *measurement oracle*. Because MGDA's QP depends on gradient magnitudes, we first normalize each gradient to unit norm, $\hat{g}_k := g_k/\|g_k\|_2$, ensuring scale-invariance across objectives with different loss magnitudes. MGDA then solves

$$\lambda^\star \in \arg\min_{\lambda \in \Delta^K} \left\| \sum_j \lambda_j \hat{g}_j \right\|^2,$$

returning weights that identify objectives locally constraining the Pareto compromise based on directional conflict rather than magnitude (App. A.5–A.6). With pairwise conflict $c_{k,j} := [-\cos(g_k, g_j)]_+$, we define interference as

$\mathrm{Conf}_k := \sum_{j \neq k} \lambda_j^\star c_{k,j}$, which is large when $k$ conflicts with several Pareto-relevant objectives.

#### 4.1.3. COMPOSITE DIFFICULTY STATE AND NORMALIZED LOSSES

Finally, we combine spectral demand and interference into a single bounded difficulty state. At each sensing step we robust-normalize $\mathrm{RQ}_k$ and $\mathrm{Conf}_k$ across objectives to obtain $\bar{R}_k$ and $\bar{C}_k$, and update

$$q_k := \alpha \bar{R}_k + \beta \bar{C}_k,$$
$$D_k \leftarrow \mathrm{clip}\left((1-\rho) D_k + \rho\, q_k, [D_{\min}, D_{\max}]\right), \quad (4)$$

where $\bar{R}_k, \bar{C}_k$ denote the normalized scores and $q_k$ is the instantaneous difficulty drive. We use this linear combination as a monotone aggregator with interpretable coefficients; App. A.7 discusses alternatives. In parallel, we maintain a normalized loss state $\tilde{L}_k$ for planning by smoothing with an EMA and dividing by a fixed scale, so the planner is invariant to loss units (see App. B).

### 4.2. Strategic Planning: Log-Hypervolume Allocation

The planner converts the state estimates from Sec. 4.1 into an epoch-level allocation target $f(t) \in \Delta^K$, where $f_k(t)$ is the desired fraction of blocks to devote to objective $k$ during

epoch $t$. We want a scalar progress signal that is *Pareto-compliant*: if all normalized losses weakly improve and at least one strictly improves (a strict Pareto improvement), the signal should strictly increase. A widely used Pareto-compliant choice is the dominated hypervolume (HV) relative to a reference point; in our setting we apply HV to the *singleton* loss vector $\tilde{\ell}(t) = (\tilde{L}_1(t), \ldots, \tilde{L}_K(t))$.

**Singleton hypervolume and reference point.** Let $\mathbf{r}(t) \in \mathbb{R}^K$ be a reference point such that $r_k(t) > \tilde{L}_k(t)$ for all $k$. For minimization, the dominated hypervolume of a single point reduces to the axis-aligned box volume $\mathrm{HV}(t) := \prod_k (r_k(t) - \tilde{L}_k(t))$. We use its log for numerical stability and separable sensitivities:

$$\phi(t) := \log \mathrm{HV}(t) = \sum_{k=1}^{K} \log \big(r_k(t) - \tilde{L}_k(t)\big). \quad (5)$$

All formal properties (Pareto compliance for fixed $\mathbf{r}$, reference-point requirements, singleton reduction) are proved in App. B. Crucially, Pareto-compliance statements hold conditional on a fixed reference point; in practice we update $\mathbf{r}(t)$ conservatively (monotone safeguard $r_k \leftarrow \max(r_k, \tilde{L}_k + \delta)$) to maintain positivity. We therefore treat $\phi(t)$ as a *priority signal* for planning within each epoch, not as a globally comparable progress metric across training.

**Log-HV sensitivities as planning priorities.** Differentiating (5) w.r.t. $\tilde{L}_k(t)$ (holding $\mathbf{r}(t)$ fixed) yields a closed-form marginal sensitivity,

$$w_k^{\mathrm{HV}}(t) := \frac{1}{r_k(t) - \tilde{L}_k(t) + \varepsilon}, \quad (6)$$

which measures how strongly improving objective $k$ would increase log-HV locally. The full derivation and the relationship between HV and log-HV gradients are provided in App. B.3.

**Allocation plan.** We convert sensitivities into an epoch-level allocation plan by tempering with the difficulty state $D_k(t)$ from Sec. 4.1. Define the difficulty-adjusted priority $a_k := w_k^{\mathrm{HV}}/(1 + \gamma D_k)$; then

$$f_k(t) := a_k(t) / \textstyle\sum_{j=1}^{K} a_j(t). \quad (7)$$

Intuitively, $w_k^{\mathrm{HV}}$ answers *which objective is most valuable for Pareto-compliant progress*, while difficulty tempering answers *how efficiently compute converts into progress*. Within epoch $t$, this induces a cumulative reference allocation $N_k^{\mathrm{ref}}(m) := f_k(t) \cdot m$ at block $m$, which serves as the target tracked by the controller in Sec. 4.3. App. B.5 derives (7) from a proportional-fair planning objective.

### 4.3. Tactical Execution: PID Tracking of the Allocation Plan

The planner outputs an epoch-level target fraction $f(t) \in \Delta^K$, but execution is discrete: each block trains exactly one task. Tactical execution therefore solves a tracking problem: within epoch $t$, choose a task sequence $\{k_{t,m}\}_{m \geq 1}$ so that the realized counts match the reference trajectory $N_k^{\mathrm{ref}}(m) := m f_k(t)$ from Sec. 4.2. We deliberately control *allocation counts* rather than losses: a scheduling decision changes counts deterministically by one, whereas losses are noisy and respond indirectly to a single block. This yields a well-conditioned feedback channel and a controller that is easy to audit via tracking plots.

**Deficits and causal tracking error.** Let $N_k(m) := \sum_{\tau=1}^{m} \mathbb{1}\{k_{t,\tau} = k\}$ be the number of blocks assigned to task $k$ after executing $m$ blocks in epoch $t$. At the *start* of block $m$ (before choosing $k_{t,m}$), the system state is $N_k(m-1)$ and the desired cumulative allocation after executing this block is $N_k^{\mathrm{ref}}(m) = m f_k(t)$. We therefore define the *pre-decision* deficit

$$e_k(m) := N_k^{\mathrm{ref}}(m) - N_k(m-1). \quad (8)$$

Selecting task $k$ at block $m$ increments $N_k$ by exactly 1 and leaves other $N_j$ unchanged, inducing a direct, stable error dynamics (formalized in App. C).

**PID logits and stochastic scheduling.** We compute per-task control logits from the deficit, its running integral, and its discrete derivative:

$$\begin{aligned} I_k(m) &:= I_k(m-1) + e_k(m), \\ \Delta e_k(m) &:= e_k(m) - e_k(m-1), \end{aligned} \quad (9)$$

$$\nu_k(m) := K_P^{(k)} e_k(m) + K_I^{(k)} I_k(m) + K_D^{(k)} \Delta e_k(m). \quad (10)$$

The logits are mapped to a sampling distribution with explicit exploration,

$$\begin{aligned} p(m) &:= (1-\epsilon) \, \mathrm{softmax}\big(\tfrac{\nu(m)}{\tau}\big) + \tfrac{\epsilon}{K} \mathbf{1}, \\ k_{t,m} &\sim \mathrm{Cat}(p(m)). \end{aligned} \quad (11)$$

Intuitively, the proportional term prioritizes tasks that are behind schedule, the integral term removes steady-state bias under fractional targets, and the derivative term damps oscillations. The $\epsilon$-exploration floor mitigates starvation and yields a geometric drought-tail bound (App. C.4). Given the selected task $k_{t,m}$, we perform the single-task block update in Eq. (2), update the count $N_k(m)$, and repeat—completing the sense–plan–control loop at block time. Theoretical properties and implementation stabilizers are given in App. C.

| Method | Cora | CiteSeer | Chameleon | Squirrel | Actor | PubMed | Wiki-CS | Co-CS | Arxiv | Avg Rank |
|---|---|---|---|---|---|---|---|---|---|---|
| BGRL | $77.14_{\pm2.2}$ | $35.28_{\pm2.1}$ | $64.87_{\pm1.3}$ | $46.11_{\pm1.2}$ | $31.22_{\pm0.5}$ | $79.94_{\pm1.2}$ | $80.36_{\pm0.3}$ | $93.05_{\pm0.4}$ | $71.24_{\pm0.8}$ | 9.2 |
| DGI | $75.64_{\pm1.6}$ | $63.84_{\pm1.6}$ | $66.27_{\pm0.9}$ | $48.17_{\pm1.0}$ | $30.46_{\pm0.7}$ | $81.28_{\pm1.1}$ | $77.95_{\pm0.9}$ | $93.17_{\pm0.4}$ | $70.86_{\pm0.6}$ | 9.6 |
| GRACE | $68.88_{\pm2.1}$ | $58.96_{\pm2.3}$ | $66.23_{\pm0.4}$ | $54.06_{\pm1.2}$ | $28.80_{\pm0.5}$ | $80.54_{\pm0.8}$ | $79.78_{\pm0.3}$ | $93.62_{\pm0.2}$ | – | 9.1 |
| MVGRL | $79.32_{\pm0.9}$ | $59.68_{\pm3.9}$ | $55.00_{\pm1.1}$ | – | $31.17_{\pm0.7}$ | $78.60_{\pm0.9}$ | – | $90.51_{\pm4.5}$ | – | 12.4 |
| p_link | $76.72_{\pm1.0}$ | $52.76_{\pm1.9}$ | $57.41_{\pm3.6}$ | $36.56_{\pm1.3}$ | $29.00_{\pm0.6}$ | $81.28_{\pm0.5}$ | $79.22_{\pm0.3}$ | $91.92_{\pm0.2}$ | $70.42_{\pm0.5}$ | 13.1 |
| p_recon | $78.76_{\pm0.5}$ | $64.86_{\pm1.5}$ | $69.39_{\pm1.3}$ | $52.33_{\pm0.8}$ | $27.75_{\pm0.4}$ | $76.28_{\pm0.9}$ | $79.88_{\pm0.4}$ | $94.85_{\pm0.3}$ | $71.08_{\pm0.4}$ | 6.8 |
| p_minsg | $76.44_{\pm1.1}$ | $61.74_{\pm0.8}$ | $64.78_{\pm1.6}$ | $47.61_{\pm0.9}$ | $29.16_{\pm0.7}$ | $80.76_{\pm1.0}$ | $79.95_{\pm0.2}$ | $93.44_{\pm0.3}$ | $70.94_{\pm0.6}$ | 10.2 |
| p_decor | $43.86_{\pm9.0}$ | $34.22_{\pm5.7}$ | $52.59_{\pm0.8}$ | $40.88_{\pm1.1}$ | $25.92_{\pm0.9}$ | $75.66_{\pm1.3}$ | $72.60_{\pm4.7}$ | $93.39_{\pm0.3}$ | $68.52_{\pm1.2}$ | 15.6 |
| p_par | $67.22_{\pm1.3}$ | $52.74_{\pm1.7}$ | $57.06_{\pm3.4}$ | $40.96_{\pm0.6}$ | $28.26_{\pm1.3}$ | $74.20_{\pm0.8}$ | $73.56_{\pm0.2}$ | $92.31_{\pm0.2}$ | $69.86_{\pm0.7}$ | 14.7 |
| AutoSSL | $80.30_{\pm1.5}$ | $64.72_{\pm1.0}$ | $70.09_{\pm2.0}$ | $49.99_{\pm3.0}$ | $29.76_{\pm0.7}$ | $80.80_{\pm0.7}$ | $79.82_{\pm0.3}$ | $93.61_{\pm0.2}$ | – | 6.4 |
| WAS | $74.12_{\pm3.3}$ | $57.66_{\pm3.5}$ | $57.59_{\pm6.3}$ | $43.52_{\pm2.2}$ | $29.68_{\pm0.5}$ | $78.62_{\pm2.1}$ | $77.40_{\pm1.6}$ | $92.62_{\pm1.3}$ | $70.68_{\pm0.9}$ | 12.9 |
| Uniform | $77.14_{\pm0.9}$ | $58.68_{\pm1.2}$ | $66.23_{\pm1.4}$ | $49.93_{\pm1.0}$ | $29.26_{\pm0.5}$ | $82.38_{\pm0.5}$ | $80.48_{\pm0.2}$ | $93.97_{\pm0.2}$ | $71.32_{\pm0.4}$ | 7.3 |
| ParetoGNN | $78.26_{\pm0.9}$ | $58.32_{\pm1.1}$ | $65.35_{\pm2.2}$ | $45.00_{\pm1.9}$ | $28.95_{\pm1.0}$ | $67.72_{\pm3.3}$ | $79.94_{\pm0.1}$ | $93.96_{\pm0.2}$ | $70.56_{\pm0.8}$ | 10.8 |
| PCGrad | $79.44_{\pm0.8}$ | $59.66_{\pm1.4}$ | $66.84_{\pm1.4}$ | $49.11_{\pm1.3}$ | $29.63_{\pm0.5}$ | $83.12_{\pm0.5}$ | $80.40_{\pm0.2}$ | $94.07_{\pm0.2}$ | $71.48_{\pm0.3}$ | 5.8 |
| CAGrad | $80.40_{\pm0.4}$ | $62.66_{\pm0.3}$ | $66.75_{\pm0.6}$ | $50.34_{\pm1.4}$ | $29.67_{\pm0.5}$ | $82.84_{\pm0.9}$ | $79.94_{\pm0.3}$ | $94.31_{\pm0.1}$ | $71.62_{\pm0.4}$ | 5.2 |
| Random | $79.06_{\pm0.3}$ | $58.10_{\pm1.3}$ | $63.07_{\pm1.8}$ | $46.22_{\pm0.7}$ | $29.57_{\pm1.0}$ | $80.56_{\pm0.8}$ | $80.10_{\pm0.2}$ | $93.39_{\pm0.2}$ | $71.18_{\pm0.5}$ | 9.4 |
| Round-Robin | $78.52_{\pm2.0}$ | $59.90_{\pm1.1}$ | $64.61_{\pm1.1}$ | $48.17_{\pm1.1}$ | $30.07_{\pm0.5}$ | $80.78_{\pm0.5}$ | $80.22_{\pm0.1}$ | $94.11_{\pm0.2}$ | $71.28_{\pm0.4}$ | 7.4 |
| ControlG | $\mathbf{81.92_{\pm0.9}}$ | $\mathbf{66.48_{\pm1.1}}$ | $69.54_{\pm1.0}$ | $53.18_{\pm0.9}$ | $31.20_{\pm0.6}$ | $\mathbf{84.24_{\pm0.7}}$ | $80.45_{\pm0.3}$ | $\mathbf{96.14_{\pm0.2}}$ | $\mathbf{72.86_{\pm0.3}}$ | **1.4** |

*Table 1.* **Node classification accuracy.** Mean ± 95% CI over 5 seeds. First , second , and third best methods per dataset are highlighted. **Avg Rank** is the mean rank across datasets (lower is better). ControlG achieves the best average rank, consistently performing among the top methods across both homophilic and heterophilic graphs.

| Variant | Cora | CiteSeer | Chameleon | Squirrel | Actor | PubMed | Wiki-CS | Co-CS | Arxiv |
|---|---|---|---|---|---|---|---|---|---|
| ControlG (full) | $\mathbf{81.92_{\pm0.9}}$ | $\mathbf{66.48_{\pm1.1}}$ | $\mathbf{69.54_{\pm1.0}}$ | $\mathbf{53.18_{\pm0.9}}$ | $\mathbf{31.20_{\pm0.6}}$ | $\mathbf{84.24_{\pm0.7}}$ | $\mathbf{80.45_{\pm0.3}}$ | $\mathbf{96.14_{\pm0.2}}$ | $\mathbf{72.86_{\pm0.3}}$ |
| w/o spectral demand ($\alpha{=}0$) | $80.14_{\pm1.2}$ | $64.72_{\pm1.4}$ | $67.82_{\pm1.3}$ | $51.64_{\pm1.1}$ | $30.42_{\pm0.8}$ | $82.68_{\pm0.9}$ | $79.12_{\pm0.5}$ | $95.42_{\pm0.3}$ | $71.94_{\pm0.5}$ |
| w/o interference ($\beta{=}0$) | $80.86_{\pm1.0}$ | $65.34_{\pm1.2}$ | $68.26_{\pm1.1}$ | $52.12_{\pm1.0}$ | $30.68_{\pm0.7}$ | $83.42_{\pm0.8}$ | $79.68_{\pm0.4}$ | $95.78_{\pm0.2}$ | $72.28_{\pm0.4}$ |
| w/o planner (uniform $\mathbf{f}$) | $78.52_{\pm1.4}$ | $62.86_{\pm1.6}$ | $65.48_{\pm1.5}$ | $49.86_{\pm1.3}$ | $29.56_{\pm0.9}$ | $80.86_{\pm1.1}$ | $77.54_{\pm0.6}$ | $94.56_{\pm0.4}$ | $70.42_{\pm0.7}$ |
| w/o controller (i.i.d. from $\mathbf{f}$) | $80.28_{\pm1.1}$ | $64.96_{\pm1.3}$ | $67.94_{\pm1.2}$ | $51.82_{\pm1.0}$ | $30.54_{\pm0.7}$ | $82.96_{\pm0.8}$ | $79.24_{\pm0.4}$ | $95.52_{\pm0.3}$ | $71.76_{\pm0.5}$ |
| w/o state signals ($\alpha{=}\beta{=}0$) | $77.86_{\pm1.5}$ | $61.92_{\pm1.7}$ | $64.72_{\pm1.6}$ | $48.94_{\pm1.4}$ | $29.18_{\pm1.0}$ | $79.94_{\pm1.2}$ | $76.82_{\pm0.7}$ | $94.18_{\pm0.5}$ | $69.82_{\pm0.8}$ |

*Table 2.* **Ablation study (node classification accuracy).** Each row removes one component from ControlG. Removing the planner or both state signals ($\alpha{=}\beta{=}0$) causes the largest drops, confirming that adaptive allocation and state estimation are essential. The interference term ($\beta$) matters most on heterophilic graphs (Chameleon, Squirrel), while spectral demand ($\alpha$) provides consistent gains.

# 5. Experiments

Our experiments are designed to test whether ControlG delivers consistent improvements over single-pretext and multi-pretext baselines while remaining computationally efficient at scale. Beyond accuracy, we also evaluate whether ControlG exhibits the behaviors it is designed for: adaptation to nonstationarity (drift), resilience to objective conflicts (disagreement), and mitigation of objective starvation (drought). To keep the main paper concise, we present the primary downstream tables in the main text and defer detailed hyperparameters, dataset statistics, ablations, and diagnostic plots to App. D.

## 5.1. Experimental setup

**Datasets.** We evaluate on nine benchmarks spanning homophilic, heterophilic, and large-scale regimes. The homophilic set includes citation networks (CORA, CITESEER, PUBMED) using Planetoid splits (Yang et al., 2016), the COAUTHOR-CS collaboration network (Shchur et al., 2018), and WIKI-CS (Mernyei & Cangea, 2020). The heterophilic set includes CHAMELEON and SQUIRREL (Rozemberczki et al., 2019) and ACTOR (Pei et al., 2020). For scalability, we include OGBN-ARXIV from OGB (Hu et al., 2020). Dataset statistics and preprocessing details are summarized in App. D.2.

**Pretext task pool (K=5).** Following task pools used in prior multi-pretext work (Jin et al., 2021; Ju et al., 2022), we pretrain with a fixed pool of five complementary objectives: link prediction (p_link), masked feature reconstruction (p_recon), node–subgraph mutual-information contrast (p_minsg), representation decorrelation (p_decor), and METIS partition prediction (p_par). Intuitively, they cover local topology, feature denoising, augmentation-invariant local context, redundancy reduction, and mesoscale community structure. We provide precise loss definitions and

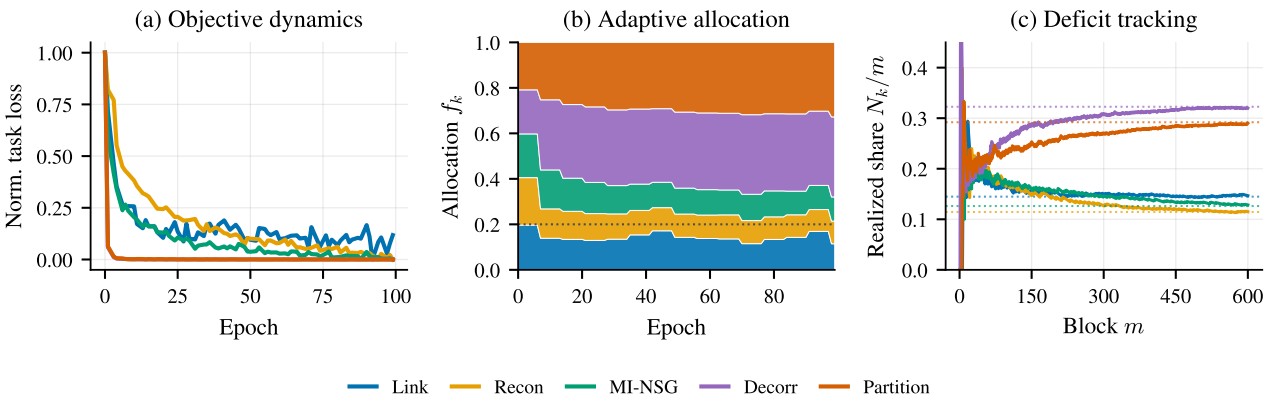

*Figure 4.* **ControlG is auditable in closed loop** (Cora). **(a) Objective dynamics:** objectives plateau at different rates—p_par, p_decor, and p_link plateau within a few epochs while p_recon and p_minsg keep improving—so per-objective utility is nonstationary (*drift*). **(b) Adaptive allocation:** the planner departs from the uniform $1/K$ target (dotted), reallocating budget across training in response to the changing state. **(c) Deficit tracking:** the PID controller drives each task's realized share $N_k/m$ to its planned allocation $f_k$ (dotted), promoting coverage and bounding the probability of starvation (*drought*).

**Algorithm 1 ControlG training loop.** Sense–plan–control scheduling: full-graph state estimation (spectral demand and interference), log-hypervolume planning, and deficit-tracking PID execution of single-task blocks.

---

**Require:** Graph $G$; objectives $\{L_k\}_{k=1}^K$; epochs $T$; block size $B_{\text{block}}$; sensing period $u$

1: **[CTRL]** Initialize $\theta$, per-task states $D_k, \tilde{L}_k, r_k$ (App. A, B)
2: **for** epoch $t = 1, \ldots, T$ **do**
3:   **[CTRL]** Reset $N_k, N_k^{\text{ref}}, I_k, e_k^{\text{prev}} \leftarrow 0$ for all $k$
4:   **for** block $m = 1, \ldots, M$ **do**
5:     **if** $m = 1$ or $(m - 1) \bmod u = 0$ **then**
6:       **[SENSE]** Compute $\text{RQ}_k, \text{Conf}_k$ on $G$; update $D_k, \tilde{L}_k$ (Sec. 4.1)
7:       **[PLAN]** Update $r_k$; compute $w_k^{\text{HV}}$, plan $f$ (Sec. 4.2)
8:     **end if**
9:     **[CTRL]** $N_k^{\text{ref}} \leftarrow N_k^{\text{ref}} + f_k$; $e_k \leftarrow N_k^{\text{ref}} - N_k$; update $I_k, \Delta e_k, \nu_k$
10:    **[CTRL]** $p \leftarrow (1 - \epsilon) \text{softmax}(\nu/\tau) + \frac{\epsilon}{K} \mathbf{1}$; sample $k_{t,m} \sim \text{Cat}(p)$
11:    **[CTRL]** Run $B_{\text{block}}$ steps on $L_{k_{t,m}}$; $N_{k_{t,m}} \leftarrow N_{k_{t,m}} + 1$; $e_k^{\text{prev}} \leftarrow e_k$
12:   **end for**
13: **end for**

---

task-specific hyperparameters in App. D.1.

**Baselines.** We use GCN (Kipf, 2016) and GAT (Veličković et al., 2017) as encoder backbones, and compare against two families of self-supervised baselines. *Single-pretext SSL*: representative graph SSL methods including DGI (Veličković et al., 2018), GRACE (You et al., 2020), MV-

GRL (Hassani & Ahmadi, 2020), and BGRL (Thakoor et al., 2021). *Multi-pretext / multi-task*: weighting-based approaches such as AUTOSSL (Jin et al., 2021) and WAS (uniform or tuned weights), and Pareto/projection-style methods including PARETOGNN (Ju et al., 2022), PCGRAD (Yu et al., 2020), and CAGRAD (Liu et al., 2021).

**Evaluation (3 downstream tasks).** Following standard multi-task graph SSL practice (Ju et al., 2022), we freeze the pretrained encoder and evaluate transfer on: **node classification** (linear probe; accuracy), **link prediction** (logistic decoder; ROC-AUC), and **node clustering** (K-means on embeddings; NMI). For link prediction, we ensure no edge leakage by removing validation/test edges from the training graph. Exact split protocols, probe hyperparameters, and evaluation settings are listed in App. D.3.

### 5.2. Main results

**Random scheduling alone provides strong baselines.** Before analyzing ControlG, we highlight a striking pattern: the *Random* scheduler—which simply samples a pretext task uniformly at random for each block—often matches or outperforms sophisticated multi-task methods. On node clustering (Table 7), Random achieves an average rank of 5.0, beating AUTOSSL, WAS, PARETOGNN, PCGRAD, and CAGRAD. Similar patterns appear on link prediction (Table 6). This suggests that *temporal separation* of conflicting objectives—even without intelligent scheduling—reduces gradient interference, since objectives never share the same gradient step in any block.

**Main comparison.** Table 1 reports node classification accuracy across nine datasets. Link prediction (Table 6) and node clustering (Table 7) are deferred to App. D.5 and

App. D.6; `ControlG` achieves the best average rank on all three tasks (1.4, 1.9, and 1.8, respectively). Several patterns emerge: *(i)* `ControlG` ranks first or second on all nine datasets for node classification, with particularly strong gains on homophilic graphs (Cora +1.5%, PubMed +1.1%, Co-CS +1.8% over the next-best method). *(ii)* On heterophilic graphs (Chameleon, Squirrel, Actor), where objective conflicts are more pronounced, `ControlG` remains competitive with the strongest single-pretext baselines while substantially outperforming weighting-based approaches like WAS. *(iii)* On the large-scale ogbn-arxiv, `ControlG` achieves 72.86%—a 1.2% gain over CAGRAD that demonstrates the method scales to large graphs.

### 5.3. Ablations and diagnostics

Table 2 isolates the contribution of each `ControlG` component. We ablate: *(i)* state estimation signals by setting $\alpha=0$ (no spectral demand) or $\beta=0$ (no interference); *(ii)* the planner by replacing log-hypervolume allocation with a uniform target; and *(iii)* the controller by replacing deficit-based tracking with i.i.d. sampling from $\mathbf{f}$. A link-prediction ablation appears in App. D.7 (Table 8).

**Key findings.** Removing the planner (uniform $\mathbf{f}$) causes the largest drop (e.g., $-3.4\%$ on Cora, $-3.4\%$ on PubMed), confirming that adaptive allocation is essential. Disabling both state signals ($\alpha=\beta=0$) yields similar degradation, showing that the planner's effectiveness depends on accurate difficulty estimates. Comparing the two signals: spectral demand ($\alpha$) provides consistent gains across all graphs, while interference ($\beta$) matters most on heterophilic datasets (Chameleon, Squirrel), where cross-task conflicts are stronger. Finally, replacing the PID controller with i.i.d. sampling from $\mathbf{f}$ degrades performance by 1–2%, validating that deficit tracking improves allocation fidelity.

**Diagnostics.** Fig. 4 makes `ControlG`'s closed-loop behavior explicit on Cora: objective losses plateau at different rates (panel a), the planner adapts allocation away from uniform in response (panel b), and the controller tracks the planned budget so every objective retains coverage (panel c). App. D.9 provides further visualizations—task-selection timelines (Fig. 6), deficit traces (Fig. 7), per-task state trajectories (Fig. 8), pairwise conflict structure (Fig. 9a), and schedule differentiation (Fig. 9b)—confirming that state estimates adapt to training dynamics and that scheduling becomes increasingly differentiated over time.

**Implementation and efficiency.** We use a fixed GNN backbone (GCN/GAT variants) across all methods; for `ControlG`, each block executes $B_{\text{block}}$ steps on a single pretext task and state estimation runs every $u$ blocks on the full graph (or a memory-efficient probe for ogbn-arxiv). Crucially, `ControlG` remains practical: its amortized overhead keeps it 5–15× faster than per-step mixing methods such as AutoSSL and ParetoGNN (Fig. 5, App. D.8). Full hyperparameters and implementation details are in App. D.4.

## 6. Conclusion

We introduced `ControlG`, a control-theoretic framework that, to the best of our knowledge, is the first to cast multi-objective graph self-supervised learning as a closed-loop scheduling problem with explicit allocation tracking. Rather than forcing every parameter update to blend signals from competing objectives—a design that leads to *Disagreement* (conflict-induced negative transfer), *Drift* (nonstationary objective utility), and *Drought* (hidden starvation of under-served objectives)—`ControlG` coordinates pretext tasks through feedback-controlled temporal allocation: estimating per-objective difficulty via spectral demand and interference signals, planning target budgets via a Pareto-aware log-hypervolume planner, and executing discrete single-task blocks with a deficit-tracking PID controller. Across nine datasets spanning homophilic, heterophilic, and large-scale regimes, `ControlG` achieves the best average rank on node classification, link prediction, and node clustering, with particularly strong gains where gradient conflicts are most pronounced. The training process is fully auditable, revealing which objectives drove learning and when—providing both performance improvements and interpretability that we hope will serve as a useful template for principled multi-task learning more broadly.

## Impact Statement

This work advances self-supervised representation learning on graphs by introducing a principled framework for coordinating multiple pretext objectives. Improved graph representations have broad applicability across scientific, industrial, and societal domains: molecular and protein embeddings can accelerate drug discovery; social network embeddings can improve community detection and misinformation identification; and the label-efficiency of self-supervised methods is valuable where annotation is expensive. The interpretability of `ControlG`'s scheduling mechanism—auditing which objectives drove learning—represents a step toward more transparent machine learning systems.

However, improved embeddings could be misused for invasive profiling, and GNNs may inherit biases from underlying graph structures. We encourage fairness audits before deployment in sensitive applications. All experiments used publicly available benchmarks, and we do not foresee dual-use concerns beyond those that already accompany graph representation learning more generally.

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

# Appendix: Table of Contents

**Appendix Contents:**

*See next page for a .*

## Notation Summary

Key notation used throughout the paper, organized by category.

| Symbol | Description | Symbol | Description |
|---|---|---|---|
| *Graph, Encoder, and Spectral Quantities* | | | |
| $G = (V, E, \mathbf{X})$ | Attributed graph | $\mathbf{A}, \mathbf{D}$ | Adjacency, degree matrices |
| $n = |V|$ | Number of nodes | $\mathbf{X} \in \mathbb{R}^{n \times d}$ | Node features |
| $f_\theta, \mathbf{Z} \in \mathbb{R}^{n \times h}$ | Encoder, embeddings | $z_v$ | Embedding of node $v$ |
| $\tilde{\mathbf{L}}$ | Normalized Laplacian | $\mathcal{E}(\mathbf{H})$ | Dirichlet energy |
| $\mathrm{RQ}_k$ | Rayleigh quotient (spectral demand) | $H_k \in \mathbb{R}^{n \times h}$ | Representation-gradient field |
| *Objectives, Losses, and Gradients* | | | |
| $K$ | Number of objectives | $\mathcal{T}_k$ | Pretext task $k$ |
| $L_k(\theta)$ | Loss (deterministic) | $\mathcal{L}_k(\theta)$ | Expected loss (stochastic) |
| $\tilde{L}_k, \tilde{\ell}$ | Normalized loss, vector | $g_k = \nabla_\theta L_k$ | Parameter gradient |
| $\hat{g}_k$ | Unit-normalized gradient | $\Delta^K$ | Probability simplex |
| $\lambda^\star, g_{\mathrm{mix}}$ | MGDA weights, mixed gradient | $\mathrm{Conf}_k$ | Interference score |
| *State Estimation and Planning* | | | |
| $D_k$ | Difficulty state | $\alpha, \beta, \gamma, \rho$ | Difficulty hyperparams |
| $\mathbf{r}(t)$ | HV reference point | $\mathrm{HV}(t), \phi(t)$ | Hypervolume, log-HV |
| $w_k^{\mathrm{HV}}$ | Log-HV sensitivity | $a_k$ | Adjusted priority |
| $f(t) \in \Delta^K$ | Allocation plan | $\delta, \varepsilon$ | Margin, stabilizer |
| *Timescales and Control* | | | |
| $t, T$ | Epoch index, total epochs | $m, M$ | Block index, blocks/epoch |
| $u$ | Sensing period | $B_{\mathrm{ep}}, B_{\mathrm{block}}$ | Batches/epoch, batches/block |
| $N_k(m)$ | Cumulative task-$k$ count | $N_k^{\mathrm{ref}}(m)$ | Reference count |
| $e_k(m)$ | Pre-decision deficit | $I_k(m), \Delta e_k$ | Integral, derivative |
| $\nu_k(m)$ | PID control logit | $K_P, K_I, K_D$ | PID gains |
| $\tau, \epsilon$ | Temperature, exploration | $k_{t,m}$ | Selected task |
| *Miscellaneous* | | | |
| $\eta$ | Learning rate | $\mathbb{1}\{\cdot\}$ | Indicator function |
| $\mathrm{clip}(x, [a, b])$ | Clipping function | $[x]_+$ | Positive part |

**Note:** We use $\tilde{\ell}$ for the normalized loss vector to distinguish from the normalized Laplacian $\tilde{\mathbf{L}}$.

# A. State Estimation: Theory and Proofs

This appendix provides formal derivations and proofs for the state-estimation quantities used in Sec. 4.1. Throughout, we assume an undirected graph unless stated otherwise so that the normalized Laplacian is symmetric PSD and admits an orthonormal eigenbasis.

## A.1. Normalized Laplacian preliminaries

Let $G = (V, E)$ be an undirected weighted graph with symmetric adjacency $\mathbf{A} \in \mathbb{R}^{n \times n}$ ($n = |V|$), degrees $d_i = \sum_j A_{ij}$, and $\mathbf{D} = \mathrm{diag}(d_1, \dots, d_n)$. The symmetric normalized Laplacian is

$$\tilde{\mathbf{L}} := \mathbf{I} - \mathbf{D}^{-1/2} \mathbf{A} \mathbf{D}^{-1/2}.$$

**Claim A.1** (Eigenvalue range of the normalized Laplacian). *All eigenvalues of $\tilde{\mathbf{L}}$ lie in $[0, 2]$.*

*Proof.* Let $\tilde{\mathbf{S}} := \mathbf{D}^{-1/2} \mathbf{A} \mathbf{D}^{-1/2}$, so $\tilde{\mathbf{L}} = \mathbf{I} - \tilde{\mathbf{S}}$. Since $\tilde{\mathbf{S}}$ is symmetric, its spectral radius equals $\|\tilde{\mathbf{S}}\|_2 = \max_{\|x\|_2 = 1} |x^\top \tilde{\mathbf{S}} x|$. For any $x \in \mathbb{R}^n$, define $y := \mathbf{D}^{-1/2} x$. Then

$$x^\top \tilde{\mathbf{S}} x = x^\top \mathbf{D}^{-1/2} \mathbf{A} \mathbf{D}^{-1/2} x = y^\top \mathbf{A} y = \sum_{i,j} A_{ij} y_i y_j.$$

Assuming no self-loops (or absorbing them into the diagonal similarly), apply the inequality $2|ab| \le a^2 + b^2$ edgewise to bound the absolute value:

$$|A_{ij} y_i y_j| \le \frac{A_{ij}}{2}(y_i^2 + y_j^2).$$

Summing over all pairs yields

$$|y^\top \mathbf{A} y| \le \sum_{i,j} |A_{ij} y_i y_j| \le \frac{1}{2} \sum_{i,j} A_{ij}(y_i^2 + y_j^2) = \sum_i \left( \sum_j A_{ij} \right) y_i^2 = \sum_i d_i y_i^2 = x^\top x.$$

Thus $|x^\top \tilde{\mathbf{S}} x| \le x^\top x$ for all $x$, which directly implies $\|\tilde{\mathbf{S}}\|_2 \le 1$. Equivalently, all eigenvalues of $\tilde{\mathbf{S}}$ lie in $[-1, 1]$. Therefore eigenvalues of $\tilde{\mathbf{L}} = \mathbf{I} - \tilde{\mathbf{S}}$ lie in $[0, 2]$. (This is also a standard spectral graph theory fact.) $\square$

## A.2. Dirichlet energy identities

**Claim A.2** (Edge-domain form of the normalized Laplacian quadratic form). *For any vector $x \in \mathbb{R}^n$,*

$$x^\top \tilde{\mathbf{L}} x = \frac{1}{2} \sum_{i,j=1}^n A_{ij} \left( \frac{x_i}{\sqrt{d_i}} - \frac{x_j}{\sqrt{d_j}} \right)^2.$$

*Proof.* Let $y := \mathbf{D}^{-1/2} x$. Then

$$x^\top \tilde{\mathbf{L}} x = x^\top x - x^\top \mathbf{D}^{-1/2} \mathbf{A} \mathbf{D}^{-1/2} x = y^\top \mathbf{D} y - y^\top \mathbf{A} y.$$

Expand each term:

$$y^\top \mathbf{D} y = \sum_{i=1}^n d_i y_i^2, \qquad y^\top \mathbf{A} y = \sum_{i,j=1}^n A_{ij} y_i y_j.$$

Use $d_i = \sum_j A_{ij}$ to rewrite $\sum_i d_i y_i^2 = \sum_{i,j} A_{ij} y_i^2$. By symmetry of $A$, also $\sum_{i,j} A_{ij} y_i^2 = \sum_{i,j} A_{ij} y_j^2$. Hence

$$\sum_i d_i y_i^2 = \frac{1}{2} \sum_{i,j} A_{ij} y_i^2 + \frac{1}{2} \sum_{i,j} A_{ij} y_j^2.$$

Therefore

$$x^\top \tilde{\mathbf{L}} x = \frac{1}{2} \sum_{i,j} A_{ij} y_i^2 + \frac{1}{2} \sum_{i,j} A_{ij} y_j^2 - \sum_{i,j} A_{ij} y_i y_j$$
$$= \frac{1}{2} \sum_{i,j} A_{ij} \left( y_i^2 + y_j^2 - 2 y_i y_j \right) = \frac{1}{2} \sum_{i,j} A_{ij} (y_i - y_j)^2.$$

Substituting $y_i = x_i / \sqrt{d_i}$ yields the claim. $\qquad\square$

**Claim A.3** (Matrix-signal extension). *For any matrix signal $H \in \mathbb{R}^{n \times h}$ with rows $H_{i,:}$,*

$$\operatorname{tr}(H^\top \tilde{\mathbf{L}} H) = \frac{1}{2} \sum_{i,j=1}^n A_{ij} \left\| \frac{H_{i,:}}{\sqrt{d_i}} - \frac{H_{j,:}}{\sqrt{d_j}} \right\|_2^2.$$

*Proof.* Write $H = [x^{(1)}, \ldots, x^{(h)}]$ by columns. Then

$$\operatorname{tr}(H^\top \tilde{\mathbf{L}} H) = \sum_{\ell=1}^h (x^{(\ell)})^\top \tilde{\mathbf{L}} x^{(\ell)}.$$

Apply Claim A.2 to each column $x^{(\ell)}$ and sum over $\ell$:

$$\sum_{\ell=1}^h \frac{1}{2} \sum_{i,j} A_{ij} \left( \frac{x_i^{(\ell)}}{\sqrt{d_i}} - \frac{x_j^{(\ell)}}{\sqrt{d_j}} \right)^2 = \frac{1}{2} \sum_{i,j} A_{ij} \sum_{\ell=1}^h \left( \frac{x_i^{(\ell)}}{\sqrt{d_i}} - \frac{x_j^{(\ell)}}{\sqrt{d_j}} \right)^2,$$

and the inner sum is exactly the squared Euclidean norm of the row difference, proving the result. $\qquad\square$

### A.3. Spectral meaning of the Rayleigh quotient

**Claim A.4** (Eigenbasis decomposition for matrix Dirichlet energy). *Let $\tilde{\mathbf{L}} = \mathbf{U} \boldsymbol{\Lambda} \mathbf{U}^\top$ with orthonormal $\mathbf{U}$ and $\boldsymbol{\Lambda} = \operatorname{diag}(\lambda_1, \ldots, \lambda_n)$. Then for any $H \in \mathbb{R}^{n \times h}$,*

$$\operatorname{tr}(H^\top \tilde{\mathbf{L}} H) = \sum_{i=1}^n \lambda_i \|\mathbf{U}_i^\top H\|_2^2, \qquad \|H\|_F^2 = \sum_{i=1}^n \|\mathbf{U}_i^\top H\|_2^2.$$

*Consequently, for $H \neq 0$ and ignoring the stabilizer,*

$$\frac{\operatorname{tr}(H^\top \tilde{\mathbf{L}} H)}{\|H\|_F^2} = \frac{\sum_{i=1}^n \lambda_i w_i}{\sum_{i=1}^n w_i}, \quad w_i := \|\mathbf{U}_i^\top H\|_2^2 \geq 0,$$

*i.e., the Rayleigh quotient is a convex combination (weighted average) of eigenvalues.*

*Proof.* Using cyclicity of trace and $\mathbf{U}^\top \mathbf{U} = \mathbf{I}$,

$$\operatorname{tr}(H^\top \tilde{\mathbf{L}} H) = \operatorname{tr}(H^\top \mathbf{U} \boldsymbol{\Lambda} \mathbf{U}^\top H) = \operatorname{tr}\big( (\mathbf{U}^\top H)^\top \boldsymbol{\Lambda} (\mathbf{U}^\top H) \big).$$

Let $\hat{H} := \mathbf{U}^\top H$ and denote its $i$-th row by $\hat{H}_{i,:} = \mathbf{U}_i^\top H$. Since $\boldsymbol{\Lambda}$ is diagonal,

$$\operatorname{tr}(\hat{H}^\top \boldsymbol{\Lambda} \hat{H}) = \sum_{i=1}^n \lambda_i \|\hat{H}_{i,:}\|_2^2 = \sum_{i=1}^n \lambda_i \|\mathbf{U}_i^\top H\|_2^2.$$

Similarly, $\|H\|_F^2 = \|\mathbf{U}^\top H\|_F^2 = \sum_i \|\mathbf{U}_i^\top H\|_2^2$ by orthonormality of $\mathbf{U}$. The Rayleigh-quotient expression follows. $\qquad\square$

## A.4. Why high Rayleigh quotient implies reduced attainable progress under low-pass attenuation

This section makes explicit the modeling assumption used in Sec. 4.1.1 and derives a rigorous upper bound connecting spectral demand to first-order progress.

**Assumption A.5** (Low-pass attenuation model for representation updates). Fix an objective $k$ and its representation-gradient field $H_k = \nabla_Z L_k(\theta; G) \in \mathbb{R}^{n \times h}$ at the current parameters $\theta$. Assume that, in a local linearization, a single update on objective $k$ induces an effective representation change of the form

$$\Delta Z \approx -\eta\, p(\tilde{\mathbf{L}})\, H_k,$$

where $p(\tilde{\mathbf{L}}) = \mathbf{U} \mathrm{diag}(p(\lambda_1), \ldots, p(\lambda_n))\mathbf{U}^\top$ is a symmetric graph spectral filter with a non-increasing frequency response $p(\lambda)$ (low-pass attenuation).

**Lemma A.6** (First-order progress under Assumption A.5). *Under Assumption A.5, the first-order change in objective $k$ satisfies*

$$L_k(\theta + \Delta\theta) \approx L_k(\theta) - \eta\, \mathrm{tr}\big(H_k^\top p(\tilde{\mathbf{L}})H_k\big),$$

*and therefore the first-order decrease magnitude is*

$$\Delta_k \;:=\; \eta\, \mathrm{tr}\big(H_k^\top p(\tilde{\mathbf{L}})H_k\big) = \eta \sum_{i=1}^n p(\lambda_i)\, \|\mathbf{U}_i^\top H_k\|_2^2.$$

*Proof.* By first-order Taylor expansion in representation space, $L_k(Z + \Delta Z) \approx L_k(Z) + \langle \nabla_Z L_k(Z), \Delta Z \rangle_F = L_k(Z) + \langle H_k, \Delta Z \rangle_F$. Substituting $\Delta Z \approx -\eta p(\tilde{\mathbf{L}})H_k$ gives

$$L_k(Z + \Delta Z) \approx L_k(Z) - \eta\, \langle H_k, p(\tilde{\mathbf{L}})H_k \rangle_F = L_k(Z) - \eta\, \mathrm{tr}\big(H_k^\top p(\tilde{\mathbf{L}})H_k\big).$$

Using the eigendecomposition of $p(\tilde{\mathbf{L}})$ and the same trace manipulation as in Claim A.4 yields the spectral sum form. □

**Lemma A.7** (High Rayleigh quotient forces mass at high frequencies). *Let $w_i := \|\mathbf{U}_i^\top H_k\|_2^2$ and $W := \sum_i w_i = \|H_k\|_F^2$. Define normalized weights $\mu_i := w_i/W$ (so $\sum_i \mu_i = 1$), and define the Rayleigh quotient without stabilizer $r := \sum_i \lambda_i \mu_i$. For any cutoff $\lambda_c \in [0, \lambda_{\max})$ with $\lambda_{\max} := \lambda_n$, let*

$$\pi_{\geq \lambda_c} := \sum_{i:\lambda_i \geq \lambda_c} \mu_i \quad \text{(fraction of gradient energy at frequencies } \geq \lambda_c\text{)}.$$

*Then*

$$\pi_{\geq \lambda_c} \;\geq\; \max\left\{0,\, \frac{r - \lambda_c}{\lambda_{\max} - \lambda_c}\right\}.$$

*Proof.* Write $r = \sum_{\lambda_i < \lambda_c} \lambda_i \mu_i + \sum_{\lambda_i \geq \lambda_c} \lambda_i \mu_i$. Since $\lambda_i \leq \lambda_c$ on the first sum and $\lambda_i \leq \lambda_{\max}$ on the second sum,

$$r \leq \lambda_c \sum_{\lambda_i < \lambda_c} \mu_i + \lambda_{\max} \sum_{\lambda_i \geq \lambda_c} \mu_i = \lambda_c(1 - \pi_{\geq \lambda_c}) + \lambda_{\max}\pi_{\geq \lambda_c}.$$

Rearranging yields $\pi_{\geq \lambda_c} \geq (r - \lambda_c)/(\lambda_{\max} - \lambda_c)$. If $r \leq \lambda_c$ the bound is nonpositive, so the trivial bound $0$ applies; combining gives the stated max form. □

**Proposition A.8** (Upper bound on attainable first-order progress decreases with spectral demand). *Under Assumption A.5 and Lemma A.6, fix any cutoff $\lambda_c \in [0, \lambda_{\max})$. Let $p(\lambda)$ be non-increasing. Define $p_0 := p(0)$ and $p_c := p(\lambda_c)$. Then for the per-energy progress ratio*

$$\bar{\Delta}_k := \frac{\Delta_k}{\eta \|H_k\|_F^2} = \sum_{i=1}^n p(\lambda_i)\mu_i,$$

*we have the bound*

$$\bar{\Delta}_k \;\leq\; p_0 - (p_0 - p_c)\, \pi_{\geq \lambda_c},$$

*and hence, in terms of the Rayleigh quotient $r = \sum_i \lambda_i \mu_i$,*

$$\bar{\Delta}_k \;\leq\; p_0 - (p_0 - p_c) \max\left\{0, \; \frac{r - \lambda_c}{\lambda_{\max} - \lambda_c}\right\}.$$

*In particular, for any fixed $\lambda_c$ and $\lambda_{\max}$, this upper bound is a non-increasing function of $r$; thus larger Rayleigh quotient implies a smaller upper bound on first-order progress per unit gradient energy under low-pass attenuation.*

*Proof.* Split the sum into low and high frequency parts:

$$\bar{\Delta}_k = \sum_{\lambda_i < \lambda_c} p(\lambda_i)\mu_i + \sum_{\lambda_i \geq \lambda_c} p(\lambda_i)\mu_i.$$

Since $p(\lambda)$ is non-increasing, for $\lambda_i < \lambda_c$ we have $p(\lambda_i) \leq p(0) = p_0$, and for $\lambda_i \geq \lambda_c$ we have $p(\lambda_i) \leq p(\lambda_c) = p_c$. Therefore,

$$\bar{\Delta}_k \leq p_0 \sum_{\lambda_i < \lambda_c} \mu_i + p_c \sum_{\lambda_i \geq \lambda_c} \mu_i = p_0(1 - \pi_{\geq \lambda_c}) + p_c \pi_{\geq \lambda_c} = p_0 - (p_0 - p_c)\pi_{\geq \lambda_c}.$$

Substituting the lower bound on $\pi_{\geq \lambda_c}$ from Lemma A.7 yields the inequality in terms of $r$. Since $(p_0 - p_c) \geq 0$ and the max term is non-decreasing in $r$, the bound is non-increasing in $r$. $\qquad\square$

*Remark* A.9 (How this supports "difficulty" in Sec. 4.1.1). For fixed $\|H_k\|_F$ and $\eta$, Proposition A.8 gives a decreasing upper bound on the attainable first-order decrease as the Rayleigh quotient increases. Thus achieving a target decrease in $L_k$ requires at least $\Omega(1/\bar{\Delta}_k)$ steps, which grows as the spectral demand (Rayleigh quotient) increases. This is the formal sense in which $\mathrm{RQ}_k$ can be used as a difficulty proxy under Assumption A.5.

### A.5. Interference and MGDA relevance

**Claim A.10** (Negative alignment implies first-order interference). *Let $g_k(\theta) = \nabla_\theta L_k(\theta; G)$. For $\theta^+ = \theta - \eta g_k(\theta)$ with $\eta > 0$,*

$$L_j(\theta^+) = L_j(\theta) - \eta \langle g_j(\theta), g_k(\theta)\rangle + o(\eta).$$

*In particular, if $\langle g_j, g_k\rangle < 0$, a descent step on $k$ increases $L_j$ to first order.*

*Proof.* This is the first-order Taylor expansion: $L_j(\theta - \eta g_k) = L_j(\theta) + \langle \nabla_\theta L_j(\theta), -\eta g_k(\theta)\rangle + o(\eta)$, and $\nabla_\theta L_j(\theta) = g_j(\theta)$. $\qquad\square$

**Theorem A.11** (Pareto stationarity and convex hull of gradients). *Fix gradients $g_1, \ldots, g_K \in \mathbb{R}^d$ at a point $\theta$. Define first-order Pareto stationarity (PS) as: there does not exist a direction $d \in \mathbb{R}^d$ such that $\langle g_k, d\rangle < 0$ for all $k$. Let $\mathcal{C} := \mathrm{conv}\{g_1, \ldots, g_K\}$. Then $\theta$ is PS if and only if $\mathbf{0} \in \mathcal{C}$.*

*Proof.* $(\Rightarrow)$ Suppose $\mathbf{0} \notin \mathcal{C}$. Since $\mathcal{C}$ is compact and convex and $\mathbf{0}$ is a point outside it, the strong separating hyperplane theorem implies there exists $d \neq 0$ and $c > 0$ such that $\langle d, g\rangle \geq c$ for all $g \in \mathcal{C}$. In particular, for each extreme point $g_k \in \mathcal{C}$ we have $\langle d, g_k\rangle \geq c > 0$, hence $\langle -d, g_k\rangle \leq -c < 0$ for all $k$. Thus $-d$ is a common descent direction, contradicting PS.

$(\Leftarrow)$ Conversely, if there exists $d$ such that $\langle g_k, d\rangle < 0$ for all $k$, then for any convex combination $g = \sum_k \lambda_k g_k$ with $\lambda \in \Delta^K$ we have $\langle g, d\rangle = \sum_k \lambda_k \langle g_k, d\rangle < 0$. In particular $\langle \mathbf{0}, d\rangle = 0$ cannot satisfy this strict inequality, so $\mathbf{0}$ cannot belong to the convex hull $\mathcal{C}$. Therefore, if $\mathbf{0} \in \mathcal{C}$, no such common descent direction exists; hence $\theta$ is PS. $\qquad\square$

**Theorem A.12** (MGDA as projection; descent unless Pareto-stationary). *Let $G = [g_1, \ldots, g_K] \in \mathbb{R}^{d \times K}$ and consider the convex program*

$$\min_{\lambda \in \Delta^K} \; \frac{1}{2}\|G\lambda\|_2^2.$$

*Let $\lambda^\star$ be an optimizer and define $g_{\mathrm{mix}} := G\lambda^\star \in \mathcal{C}$. Then: (i) $g_{\mathrm{mix}}$ is the Euclidean projection of $\mathbf{0}$ onto the convex hull $\mathcal{C}$; (ii) if $g_{\mathrm{mix}} = \mathbf{0}$ then $\mathbf{0} \in \mathcal{C}$ and $\theta$ is Pareto-stationary (Theorem A.11); (iii) if $g_{\mathrm{mix}} \neq \mathbf{0}$ then $d := -g_{\mathrm{mix}}$ is a common descent direction and moreover*

$$\langle g_k, d\rangle \leq -\|g_{\mathrm{mix}}\|_2^2 \quad \text{for all } k.$$

*Proof.* (i) The set $\mathcal{C} = \{G\lambda : \lambda \in \Delta^K\}$ is exactly the feasible set of convex combinations. Minimizing $\|G\lambda\|_2^2/2$ is equivalent to minimizing $\|g\|_2^2/2$ over $g \in \mathcal{C}$, hence $g_{\mathrm{mix}}$ is the closest point in $\mathcal{C}$ to the origin, i.e., the projection.

(ii) If $g_{\mathrm{mix}} = \mathbf{0}$, then $\mathbf{0} \in \mathcal{C}$ by definition of $\mathcal{C}$, and PS follows from Theorem A.11.

(iii) Projection onto a closed convex set satisfies the variational inequality: for all $g \in \mathcal{C}$, $\langle g - g_{\mathrm{mix}}, \mathbf{0} - g_{\mathrm{mix}} \rangle \leq 0$. Substituting $\mathbf{0} - g_{\mathrm{mix}} = -g_{\mathrm{mix}}$ gives

$$\langle g - g_{\mathrm{mix}}, g_{\mathrm{mix}} \rangle \geq 0 \quad \text{for all } g \in \mathcal{C}.$$

In particular, for each extreme point $g = g_k$, we obtain $\langle g_k, g_{\mathrm{mix}} \rangle \geq \langle g_{\mathrm{mix}}, g_{\mathrm{mix}} \rangle = \|g_{\mathrm{mix}}\|_2^2$. Therefore $\langle g_k, -g_{\mathrm{mix}} \rangle \leq -\|g_{\mathrm{mix}}\|_2^2$ for all $k$, proving $-g_{\mathrm{mix}}$ is a common descent direction with the stated margin. $\square$

### A.6. Why MGDA weights identify locally relevant objectives

**Proposition A.13** (KKT characterization of MGDA weights). *Let $\lambda^\star$ solve $\min_{\lambda \in \Delta^K} \frac{1}{2}\|G\lambda\|_2^2$ and $g_{\mathrm{mix}} = G\lambda^\star$. Then there exists a scalar $\nu$ such that for every $k$,*

$$\langle g_k, g_{\mathrm{mix}} \rangle \geq \|g_{\mathrm{mix}}\|_2^2,$$

*with equality for all $k$ in the active set $\mathcal{A} := \{k : \lambda_k^\star > 0\}$. Equivalently, active gradients are exactly those lying on the supporting face of $\mathcal{C}$ orthogonal to $g_{\mathrm{mix}}$.*

*Proof.* Consider the Lagrangian

$$\mathcal{L}(\lambda, \mu, \nu) = \frac{1}{2}\|G\lambda\|_2^2 - \mu^\top \lambda + \nu(\mathbf{1}^\top \lambda - 1),$$

with multipliers $\mu \succeq 0$ for $\lambda \succeq 0$ and $\nu \in \mathbb{R}$ for $\mathbf{1}^\top \lambda = 1$. Stationarity gives

$$\nabla_\lambda \mathcal{L}(\lambda^\star, \mu^\star, \nu^\star) = G^\top G\lambda^\star - \mu^\star + \nu^\star \mathbf{1} = \mathbf{0}.$$

Since $G\lambda^\star = g_{\mathrm{mix}}$, the $k$-th component reads

$$\langle g_k, g_{\mathrm{mix}} \rangle + \nu^\star = \mu_k^\star.$$

Complementary slackness gives $\mu_k^\star \lambda_k^\star = 0$. Thus, for $k \in \mathcal{A}$ we have $\lambda_k^\star > 0$ and hence $\mu_k^\star = 0$, implying $\langle g_k, g_{\mathrm{mix}} \rangle = -\nu^\star$ (a constant over $k \in \mathcal{A}$). For $k \notin \mathcal{A}$, $\mu_k^\star \geq 0$ implies $\langle g_k, g_{\mathrm{mix}} \rangle \geq -\nu^\star$.

To identify the constant, multiply the stationarity equation by $\lambda^\star$:

$$(\lambda^\star)^\top G^\top G\lambda^\star - (\lambda^\star)^\top \mu^\star + \nu^\star (\lambda^\star)^\top \mathbf{1} = 0.$$

We have $(\lambda^\star)^\top G^\top G\lambda^\star = \|G\lambda^\star\|_2^2 = \|g_{\mathrm{mix}}\|_2^2$, $(\lambda^\star)^\top \mathbf{1} = 1$, and $(\lambda^\star)^\top \mu^\star = 0$ by complementary slackness. Hence $\|g_{\mathrm{mix}}\|_2^2 + \nu^\star = 0$, i.e., $-\nu^\star = \|g_{\mathrm{mix}}\|_2^2$. Substitute back to obtain $\langle g_k, g_{\mathrm{mix}} \rangle \geq \|g_{\mathrm{mix}}\|_2^2$ for all $k$, with equality for $k \in \mathcal{A}$. The supporting-face interpretation follows because the hyperplane $\{g : \langle g, g_{\mathrm{mix}} \rangle = \|g_{\mathrm{mix}}\|_2^2\}$ supports $\mathcal{C}$ at the projection point $g_{\mathrm{mix}}$. $\square$

### A.7. Composite difficulty aggregation

**Claim A.14** (Monotonicity and boundedness of the composite difficulty update). *Assume $\alpha, \beta \geq 0$ and define the instantaneous drive term $q_k := \alpha \overline{\mathrm{RQ}}_k + \beta \overline{\mathrm{Conf}}_k$ (denoted $\bar{R}_k, \bar{C}_k$ in Eq. (4)). Then $q_k$ is coordinatewise monotone in $(\overline{\mathrm{RQ}}_k, \overline{\mathrm{Conf}}_k)$, and the update*

$$D_k \leftarrow \mathrm{clip}\left((1-\rho)D_k + \rho q_k, [D_{\min}, D_{\max}]\right)$$

*ensures $D_k \in [D_{\min}, D_{\max}]$ for all time.*

*Proof.* Monotonicity follows because $q_k$ is an affine function with nonnegative coefficients. Boundedness follows from the definition of $\mathrm{clip}(\cdot, [D_{\min}, D_{\max}])$, which maps any input to the closed interval $[D_{\min}, D_{\max}]$; thus by induction $D_k$ always lies in that interval. $\square$

# B. Log-Hypervolume Planning: Definitions and Proofs

This appendix provides formal definitions and proofs for Sec. 4.2. We consider a minimization setting throughout. Let $\tilde{\ell} = (\tilde{L}_1, \ldots, \tilde{L}_K) \in \mathbb{R}^K$ denote the normalized loss vector and let $\mathbf{r} \in \mathbb{R}^K$ be a reference point satisfying $r_k > \tilde{L}_k$ for all $k$. (Note: we use $\tilde{\ell}$ for the loss vector to avoid confusion with the normalized Laplacian $\tilde{\mathbf{L}}$ from Sec. 4.1.)

## B.1. Preliminaries: dominance and hypervolume

We first establish notation for Pareto dominance, then define the dominated hypervolume.

**Definition B.1** (Dominance for minimization). For vectors $\mathbf{a}, \mathbf{b} \in \mathbb{R}^K$, we say $\mathbf{a}$ *weakly dominates* $\mathbf{b}$ (written $\mathbf{a} \preceq \mathbf{b}$) if $a_k \leq b_k$ for all $k \in \{1, \ldots, K\}$. We say $\mathbf{a}$ *strictly dominates* $\mathbf{b}$ (written $\mathbf{a} \prec \mathbf{b}$) if $\mathbf{a} \preceq \mathbf{b}$ and $\mathbf{a} \neq \mathbf{b}$. For reference-point conditions, we write $\mathbf{r} \succ \tilde{\ell}$ to mean $r_k > \tilde{L}_k$ for all $k$ (componentwise strict inequality).

**Definition B.2** (Dominated hypervolume). Given a set of objective vectors $\mathcal{P} \subset \mathbb{R}^K$ and a reference point $\mathbf{r}$, the dominated region is

$$\mathcal{Z}(\mathcal{P}, \mathbf{r}) := \{ z \in \mathbb{R}^K \ : \ \exists y \in \mathcal{P} \text{ such that } y \preceq z \preceq \mathbf{r} \},$$

and the hypervolume indicator is the $K$-dimensional Lebesgue measure $\mathrm{HV}(\mathcal{P}, \mathbf{r}) := \mathrm{Vol}(\mathcal{Z}(\mathcal{P}, \mathbf{r}))$.

**Claim B.3** (Singleton reduction). *If $\mathcal{P} = \{\tilde{\ell}\}$ is a singleton and $\mathbf{r} \succ \tilde{\ell}$ coordinatewise, then*

$$\mathrm{HV}(\{\tilde{\ell}\}, \mathbf{r}) = \prod_{k=1}^{K} (r_k - \tilde{L}_k).$$

*Proof.* For a singleton, the dominated region is exactly the axis-aligned hyperrectangle $\mathcal{Z}(\{\tilde{\ell}\}, \mathbf{r}) = \{z \in \mathbb{R}^K : \tilde{L}_k \leq z_k \leq r_k \ \forall k\}$. Its Lebesgue measure is the product of side lengths, giving the stated expression. $\square$

## B.2. Pareto compliance for fixed reference point

The key property we require is that strict Pareto improvements yield strict increases in the planning signal.

**Theorem B.4** (Singleton HV is strictly Pareto-compliant). *Fix a reference point $\mathbf{r}$ such that $r_k > a_k$ and $r_k > b_k$ for all $k$. If $\mathbf{a}$ strictly dominates $\mathbf{b}$ (Definition B.1), then $\mathrm{HV}(\{\mathbf{a}\}, \mathbf{r}) > \mathrm{HV}(\{\mathbf{b}\}, \mathbf{r})$.*

*Proof.* By Claim B.3, we have $\mathrm{HV}(\{\mathbf{a}\}, \mathbf{r}) = \prod_{k=1}^{K} (r_k - a_k)$ and similarly for $\mathbf{b}$. Since $\mathbf{a} \preceq \mathbf{b}$, each factor satisfies $r_k - a_k \geq r_k - b_k \geq 0$. Moreover, $\mathbf{a} \neq \mathbf{b}$ implies strict inequality $r_j - a_j > r_j - b_j$ for at least one index $j$. Since all factors are strictly positive (by the reference-point condition), the product over $k$ is strictly larger for $\mathbf{a}$ than for $\mathbf{b}$. $\square$

**Corollary B.5** (Log-HV is also Pareto-compliant (fixed $\mathbf{r}$)). *Under the conditions of Theorem B.4, $\phi(\mathbf{x}) := \sum_{k=1}^{K} \log(r_k - x_k)$ satisfies $\phi(\mathbf{a}) > \phi(\mathbf{b})$.*

*Proof.* $\log(\cdot)$ is strictly increasing on $\mathbb{R}_{>0}$, so $\log \mathrm{HV}(\{\mathbf{a}\}, \mathbf{r}) > \log \mathrm{HV}(\{\mathbf{b}\}, \mathbf{r})$. Using Claim B.3 gives $\log \mathrm{HV}(\{\mathbf{x}\}, \mathbf{r}) = \sum_k \log(r_k - x_k)$. $\square$

## B.3. Log-hypervolume sensitivities

**Claim B.6** (Gradient of log-HV and the sensitivity weight). *Fix $\mathbf{r}$ and define $\phi(\tilde{\ell}) = \sum_{k=1}^{K} \log(r_k - \tilde{L}_k)$. Then for each $k$,*

$$\frac{\partial \phi}{\partial \tilde{L}_k} = -\frac{1}{r_k - \tilde{L}_k}.$$

*Consequently, the sensitivity magnitude used in Sec. 4.2,*

$$w_k^{\mathrm{HV}} := \frac{1}{r_k - \tilde{L}_k + \varepsilon},$$

*is a stabilized version of $-\partial \phi / \partial \tilde{L}_k$.*

*Proof.* The function $\phi$ is separable across coordinates. Differentiating $\log(r_k - \tilde{L}_k)$ w.r.t. $\tilde{L}_k$ yields $\partial/\partial\tilde{L}_k \log(r_k - \tilde{L}_k) = -(r_k - \tilde{L}_k)^{-1}$. All other terms do not depend on $\tilde{L}_k$. $\qquad\square$

**Claim B.7** (Relation to the HV gradient). *Let* $\mathrm{HV}(\tilde{\ell}) = \prod_k (r_k - \tilde{L}_k)$ *with fixed* $\mathbf{r}$. *Then*

$$\frac{\partial\,\mathrm{HV}}{\partial\tilde{L}_k} = -\frac{\mathrm{HV}(\tilde{\ell})}{r_k - \tilde{L}_k}, \qquad \frac{\partial\,\log\mathrm{HV}}{\partial\tilde{L}_k} = -\frac{1}{r_k - \tilde{L}_k}.$$

*Proof.* Differentiate the product: $\partial\mathrm{HV}/\partial\tilde{L}_k = -\prod_{j\neq k}(r_j - \tilde{L}_j)$, and note that $\prod_{j\neq k}(r_j - \tilde{L}_j) = \mathrm{HV}/(r_k - \tilde{L}_k)$. The second identity follows from Claim B.6 and $\log\mathrm{HV} = \sum_k \log(r_k - \tilde{L}_k)$. $\qquad\square$

*Remark* B.8 (Why log-HV sensitivities are preferred). Claim B.7 reveals a key difference: the HV gradient at coordinate $k$ scales with the current hypervolume value, whereas the log-HV gradient depends only on the slack $(r_k - \tilde{L}_k)$. Using log-HV sensitivities therefore yields priorities that are invariant to the absolute scale of the dominated region, making them more stable across training phases where HV may span orders of magnitude.

## B.4. Reference point specification and positivity

**Claim B.9** (Positivity condition for log-HV). $\phi(\tilde{\ell}) = \sum_k \log(r_k - \tilde{L}_k)$ *is well-defined if and only if* $r_k - \tilde{L}_k > 0$ *for all* $k$.

*Proof.* Each term $\log(r_k - \tilde{L}_k)$ is defined if and only if its argument is positive. The claim follows by separability. $\qquad\square$

*Remark* B.10 (Reference point initialization). To ensure positivity throughout training, we initialize $r_k$ from a short warm-up window observed on the full graph: $r_k \leftarrow \max_{t \leq t_{\mathrm{warm}}} \tilde{L}_k(t) + \delta$, matching the additive margin used in Algorithm 2. We also apply a monotone safeguard $r_k \leftarrow \max(r_k, \tilde{L}_k + \delta)$ to handle rare upward fluctuations. Since HV values depend on $\mathbf{r}$, we treat log-HV as a priority signal for planning rather than a globally comparable metric across time.

## B.5. Deriving the allocation plan

Sec. 4.2 converts per-objective planning priorities into an allocation fraction $f \in \Delta^K$. Let

$$a_k := \frac{w_k^{\mathrm{HV}}}{1 + \gamma D_k},$$

which we interpret as a difficulty-adjusted marginal value score (higher is better).

**Proposition B.11** (Proportional-fair allocation yields normalized priorities). *Consider the concave planning objective*

$$\max_{f\in\Delta^K,\, f\succ 0} \sum_{k=1}^{K} a_k \log f_k.$$

*Its unique maximizer is* $f_k^\star = a_k / \sum_{j=1}^{K} a_j$, *which matches Eq.* (7) *in Sec.* 4.2.

*Proof.* Form the Lagrangian $\mathcal{L}(f,\nu) = \sum_k a_k \log f_k + \nu(1 - \sum_k f_k)$. Stationarity gives $\partial\mathcal{L}/\partial f_k = a_k/f_k - \nu = 0$, so $f_k = a_k/\nu$. Enforcing $\sum_k f_k = 1$ yields $\nu = \sum_j a_j$, hence $f_k^\star = a_k / \sum_j a_j$. Strict concavity of $\sum_k a_k \log f_k$ on $f \succ 0$ implies uniqueness. $\qquad\square$

*Remark* B.12 (Interpretation of the proportional-fair planner). Proposition B.11 provides a principled rationale for converting priorities into a non-degenerate allocation over tasks: it favors high-priority tasks while avoiding collapsing all budget onto a single objective. Other regularizers (e.g., entropy or minimum-floor constraints) can be incorporated via standard convex analysis; we evaluate these variants in ablations.

**Allocation floor.** In practice we apply a small allocation floor $f_{\min} \in [0, 1/K)$ to the planner output, guaranteeing a minimum scheduled share per objective:

$$\tilde{f}_k = f_{\min} + (1 - K f_{\min}) \frac{a_k}{\sum_j a_j}, \qquad \text{so that} \qquad \sum_k \tilde{f}_k = 1, \quad \tilde{f}_k \geq f_{\min}. \tag{12}$$

Setting $f_{\min} = 0$ recovers the unfloored plan $f_k^\star$ of Proposition B.11. The tuned values (App. D.4, typically $f_{\min} \in \{0, 0.05, 0.1\}$) act as an implementation stabilizer that complements the controller's $\epsilon$-exploration floor; it is a strategic-level guarantee and is not required by any theorem in this section.

## C. Tactical Execution: Deficit Dynamics and Guarantees

This appendix formalizes the controller in Sec. 4.3. We fix an epoch $t$ and suppress $(t)$ in $f_k(t)$ for readability.

### C.1. Deficit dynamics and basic identities

**Definition C.1** (Counts, reference, and pre-decision deficits). Let $f \in \Delta^K$ be fixed within an epoch. Let $N_k(m) := \sum_{\tau=1}^m \mathbb{1}\{k_{t,\tau} = k\}$ be realized counts after $m$ executed blocks, and let $N_k^{\text{ref}}(m) := m f_k$ be the reference trajectory. Define the pre-decision deficit at block $m$ as

$$e_k(m) := N_k^{\text{ref}}(m) - N_k(m-1) = m f_k - N_k(m-1).$$

**Claim C.2** (Sum of deficits is constant). *For every $m \geq 1$, $\sum_{k=1}^K e_k(m) = 1$.*

*Proof.* We have

$$\sum_{k=1}^K e_k(m) = \sum_{k=1}^K \big( m f_k - N_k(m-1) \big) = m \sum_{k=1}^K f_k - \sum_{k=1}^K N_k(m-1).$$

Because $f \in \Delta^K$, $\sum_k f_k = 1$, and because exactly one task is selected per block, $\sum_k N_k(m-1) = m-1$. Hence $\sum_k e_k(m) = m - (m-1) = 1$. $\square$

**Claim C.3** (Deficit recursion). *Let $k_{t,m}$ be the task executed at block $m$. Then for every $k$ and $m \geq 1$,*

$$e_k(m+1) = e_k(m) + f_k - \mathbb{1}\{k_{t,m} = k\}.$$

*Proof.* By definition,

$$e_k(m+1) = (m+1)f_k - N_k(m) = m f_k - N_k(m-1) + f_k - \big( N_k(m) - N_k(m-1) \big).$$

The increment $N_k(m) - N_k(m-1)$ equals $\mathbb{1}\{k_{t,m} = k\}$, giving the recursion. $\square$

### C.2. A worst-case tracking guarantee in the max-deficit limit

The following result provides a clean, worst-case tracking guarantee for a canonical deterministic controller. It is the discrete analogue of deficit/credit-based fair schedulers.

**Definition C.4** (Max-deficit controller). At each block $m$, choose

$$k_{t,m} \in \arg \max_{k \in [K]} e_k(m).$$

(Ties may be broken arbitrarily.)

**Theorem C.5** (Bounded tracking error under max-deficit scheduling). *Define the post-execution discrepancy*

$$d_k(m) := N_k(m) - m f_k.$$

*Under the max-deficit controller (Def. C.4), for every $m \geq 0$ and every $k \in [K]$,*

$$d_k(m) < 1 \qquad \text{and} \qquad d_k(m) > -(K-1),$$

*so $|d_k(m)| < K - 1$. The bound is $O(K)$, and hence $O(1)$ for a fixed task pool.*

*Proof.* We proceed in two parts.

**(Upper bound: $d_k(m) < 1$).** We prove by induction on $m$. The base case $m = 0$ satisfies $d_k(0) = 0 < 1$. Assume $d_k(m-1) < 1$ for all $k$.

If task $k$ is *not* chosen at block $m$, then $N_k(m) = N_k(m-1)$ and

$$d_k(m) = N_k(m-1) - mf_k = \big(N_k(m-1) - (m-1)f_k\big) - f_k$$
$$= d_k(m-1) - f_k < d_k(m-1) < 1.$$

If task $k$ *is* chosen at block $m$, then $N_k(m) = N_k(m-1) + 1$ and

$$d_k(m) = N_k(m-1) + 1 - mf_k = d_k(m-1) + 1 - f_k.$$

To show this is $< 1$, we use the selection rule. Note that

$$e_k(m) = mf_k - N_k(m-1) = f_k - d_k(m-1).$$

By Claim C.2, $\sum_j e_j(m) = 1$, hence $\max_j e_j(m) \geq 1/K > 0$. Since $k$ is chosen among maximizers, $e_k(m) > 0$, implying $d_k(m-1) < f_k$. Therefore

$$d_k(m) = d_k(m-1) + 1 - f_k < f_k + 1 - f_k = 1.$$

**(Lower bound: $d_k(m) > -(K-1)$).** The discrepancies sum to zero: since exactly one task is selected per block, $\sum_k N_k(m) = m$, and as $\sum_k f_k = 1$,

$$\sum_k d_k(m) = \sum_k N_k(m) - m \sum_k f_k = m - m = 0.$$

Fix any $j \in [K]$. Applying the upper bound $d_k(m) < 1$ to each of the other $K-1$ tasks,

$$d_j(m) = -\sum_{k \neq j} d_k(m) > -\sum_{k \neq j} 1 = -(K-1).$$

Combining the two bounds gives $-(K-1) < d_k(m) < 1$, i.e. $|d_k(m)| < K - 1$. $\qquad\square$

*Remark* C.6 (Why the lower bound scales with $K$). The symmetric statement $|d_k(m)| < 1$ does not hold in general for $K > 2$. Under a skewed target $f$, the max-deficit rule can repeatedly favor a few high-share tasks while a low-share task accumulates discrepancy below $-1$ before becoming the maximizer; a short deterministic simulation already exhibits $\min_{k,m} d_k(m) \approx -1.23$ for $K = 8$. The one-sided over-allocation bound $d_k(m) < 1$ is always tight, while under-allocation is controlled at the $O(K)$ scale above. In ControlG the $\epsilon$-exploration floor (Sec. C.4) provides a complementary, allocation-independent visitation guarantee.

### C.3. Relation to the PID-softmax controller

**Claim C.7** (Softmax concentrates on argmax as temperature vanishes). *Fix $u \in \mathbb{R}^K$ and define $q_\tau = \mathrm{softmax}(u/\tau)$ with $\tau > 0$. Let $\mathcal{A} = \arg\max_k u_k$. Then as $\tau \downarrow 0$, $q_\tau$ assigns probability mass only to $\mathcal{A}$, and for any $i \notin \mathcal{A}$, $q_{\tau,i} \to 0$.*

*Proof.* Let $u^\star = \max_k u_k$. For any $i$,

$$q_{\tau,i} = \frac{e^{u_i/\tau}}{\sum_j e^{u_j/\tau}} = \frac{e^{(u_i - u^\star)/\tau}}{\sum_j e^{(u_j - u^\star)/\tau}}.$$

If $i \notin \mathcal{A}$ then $u_i - u^\star < 0$, so $e^{(u_i - u^\star)/\tau} \to 0$ as $\tau \downarrow 0$, while for $j \in \mathcal{A}$ the exponent is $0$ and the corresponding terms remain $1$. Hence $q_{\tau,i} \to 0$ for $i \notin \mathcal{A}$ and total mass concentrates on $\mathcal{A}$. $\qquad\square$

*Remark* C.8 (Max-deficit as a limiting case of PID-softmax). If $K_I^{(k)} = K_D^{(k)} = 0$, $\epsilon = 0$, and $\nu_k(m) = K_P e_k(m)$, then $\arg\max_k \nu_k(m) = \arg\max_k e_k(m)$ and by Claim C.7 the sampling rule in Eq. (11) approaches the deterministic max-deficit controller as $\tau \downarrow 0$.

## C.4. Starvation prevention under explicit exploration

**Claim C.9** (Uniform lower bound on per-step selection probability). *Under Eq. (11), for every $m$ and every $k$, $p_k(m) \geq \epsilon/K$.*

*Proof.* Eq. (11) is a convex combination of two distributions: $(1 - \epsilon) \operatorname{softmax}(\cdot)$ and $\epsilon \cdot \mathbf{1}/K$. Therefore each coordinate is at least $\epsilon/K$. ☐

**Proposition C.10** (Geometric tail bound on drought length). *Fix any task $k$ and any time $m_0$. Let $T$ be the first time $m > m_0$ such that $k_{t,m} = k$. Then for any integer $s \geq 1$,*

$$\Pr\big(T > m_0 + s \,\big|\, \mathcal{F}_{m_0}\big) \ \leq \ \Big(1 - \frac{\epsilon}{K}\Big)^s,$$

*where $\mathcal{F}_{m_0}$ is the sigma-field generated by the history up to time $m_0$.*

*Proof.* By Claim C.9, for every $m \geq m_0$ we have

$$\Pr(k_{t,m} \neq k \mid \mathcal{F}_{m-1}) = 1 - p_k(m) \leq 1 - \epsilon/K.$$

Therefore, using the chain rule for conditional probabilities,

$$\begin{aligned}
\Pr(T > m_0 + s \mid \mathcal{F}_{m_0}) &= \Pr \Big( \bigcap_{r=1}^{s} \{k_{t,m_0+r} \neq k\} \,\Big|\, \mathcal{F}_{m_0} \Big) \\
&= \prod_{r=1}^{s} \Pr(k_{t,m_0+r} \neq k \mid \mathcal{F}_{m_0+r-1}) \\
&\leq \Big(1 - \frac{\epsilon}{K}\Big)^s.
\end{aligned}$$

☐

**Corollary C.11** (Bound on expected waiting time). *Under the conditions of Prop. C.10, $\mathbb{E}[T - m_0 \mid \mathcal{F}_{m_0}] \leq K/\epsilon$.*

*Proof.* A nonnegative integer-valued random variable satisfies $\mathbb{E}[X] = \sum_{s \geq 0} \Pr(X > s)$. Apply Prop. C.10 with $X = T - m_0$:

$$\begin{aligned}
\mathbb{E}[T - m_0 \mid \mathcal{F}_{m_0}] &= \sum_{s \geq 0} \Pr(T > m_0 + s \mid \mathcal{F}_{m_0}) \\
&\leq \sum_{s \geq 0} \Big(1 - \frac{\epsilon}{K}\Big)^s = \frac{K}{\epsilon}.
\end{aligned}$$

☐

## C.5. Implementation stabilizers

**Claim C.12** (Anti-windup by clipping keeps the integral state bounded). *If the integrator update is implemented as*

$$I_k(m) \leftarrow \operatorname{clip}\big(I_k(m - 1) + e_k(m), [-I_{\max}, I_{\max}]\big),$$

*then $|I_k(m)| \leq I_{\max}$ for all $m$.*

*Proof.* Immediate from the definition of $\operatorname{clip}(\cdot, [-I_{\max}, I_{\max}])$. ☐

*Remark* C.13 (Why anti-windup is necessary). When the controller output is effectively saturated (here, logits/probabilities cannot instantaneously realize arbitrary reference changes), the integral term can accumulate persistent error and degrade responsiveness. Clipping (or back-calculation) is a standard anti-windup safeguard in PID control.

# D. Additional Experimental Details

## D.1. Pretext task objectives

Let $G = (V, E, X)$ be a graph with $|V| = n$ nodes, adjacency $A$, and node features $X \in \mathbb{R}^{n \times d_x}$. The encoder $f_\theta$ maps the graph (or an augmented view of it) to node embeddings $Z = f_\theta(G) \in \mathbb{R}^{n \times d}$, with row vector $z_v \in \mathbb{R}^d$ for node $v$. Unless otherwise stated, objectives are estimated on minibatches (or sampled subgraphs) but are defined below at the population level for clarity.

**(p_link) Link prediction (LP).** We train a decoder $s_\phi(z_u, z_v)$ (e.g., dot-product or an MLP) to predict whether an edge exists between node pairs. Let $E^+$ denote observed (positive) edges and $E^-$ denote sampled non-edges (negative pairs). The link prediction objective is a binary cross-entropy loss:

$$\mathcal{L}_{\mathrm{LP}}(\theta, \phi) = -\frac{1}{|E^+|} \sum_{(u,v) \in E^+} \log \sigma(s_\phi(z_u, z_v)) - \frac{1}{|E^-|} \sum_{(u,v) \in E^-} \log(1 - \sigma(s_\phi(z_u, z_v))), \tag{13}$$

where $\sigma(\cdot)$ is the logistic sigmoid. When downstream evaluation includes link prediction, we compute $\mathcal{L}_{\mathrm{LP}}$ on the *training* graph only (validation/test edges removed) to prevent leakage.

**(p_recon) Masked feature reconstruction (MFR).** We randomly mask a subset of nodes (or features) and train the model to reconstruct the original attributes. Let $m \in \{0, 1\}^n$ indicate which nodes are masked and let $\tilde{X} = \mathrm{Mask}(X; m)$ denote corrupted features (e.g., replaced by a mask token or zeros). We encode the corrupted graph to get $\tilde{Z} = f_\theta(G, \tilde{X})$ and decode features with $g_\psi$: $\hat{X} = g_\psi(\tilde{Z})$. We reconstruct only masked entries:

$$\mathcal{L}_{\mathrm{MFR}}(\theta, \psi) = \frac{1}{\sum_v m_v} \sum_{v \in V : m_v = 1} \ell_{\mathrm{rec}}(\hat{x}_v, x_v), \tag{14}$$

where $\ell_{\mathrm{rec}}$ is a reconstruction loss (e.g., MSE for continuous features, or cross-entropy for one-hot features). Unless otherwise tuned, we mask a fraction $0.5$ of nodes and use a scaled feature-error loss for $\ell_{\mathrm{rec}}$.

**(p_minsg) Mutual-information node–subgraph contrast (MI-NSG).** This objective encourages agreement between an anchor node representation and a summary of its local subgraph context under stochastic augmentations. We generate two augmented views $G^{(1)} = \mathrm{Aug}(G)$ and $G^{(2)} = \mathrm{Aug}(G)$ (e.g., feature/edge dropout). Let $Z^{(1)} = f_\theta(G^{(1)})$ and $Z^{(2)} = f_\theta(G^{(2)})$. For each node $v$, define a local subgraph summary (context) vector

$$c_v^{(2)} = \mathrm{READOUT}(\{z_u^{(2)} : u \in \mathcal{S}(v)\}), \tag{15}$$

where $\mathcal{S}(v)$ denotes the nodes in $v$'s local neighborhood/subgraph (e.g., $r$-hop ego graph) and READOUT is a permutation-invariant pooling operator. We then optimize an InfoNCE-style contrastive objective between $z_v^{(1)}$ and its matched context $c_v^{(2)}$:

$$\mathcal{L}_{\mathrm{MI\text{-}NSG}}(\theta) = -\frac{1}{n} \sum_{v \in V} \log \frac{\exp(\mathrm{sim}(z_v^{(1)}, c_v^{(2)})/\tau)}{\sum_{u \in V} \exp(\mathrm{sim}(z_v^{(1)}, c_u^{(2)})/\tau)}, \tag{16}$$

where $\mathrm{sim}(\cdot, \cdot)$ is a similarity function (e.g., dot product or cosine) and $\tau$ is a temperature. By default we use temperature $\tau = 0.2$, a 3-hop ego-graph for $\mathcal{S}(v)$, and feature/edge dropout of $0.3$ for the augmentations $\mathrm{Aug}(\cdot)$.

**(p_decor) Representation decorrelation (RepDecor).** RepDecor reduces redundancy between embedding dimensions using a Barlow-Twins-style objective. Using two augmented views as above, we obtain embeddings for a minibatch of nodes $\mathcal{B} \subseteq V$ and a projection head $h$: $Y^{(1)} = \mathrm{Norm}(h(Z_\mathcal{B}^{(1)}))$ and $Y^{(2)} = \mathrm{Norm}(h(Z_\mathcal{B}^{(2)}))$. Let $C \in \mathbb{R}^{d \times d}$ be the cross-correlation matrix:

$$C_{ij} = \frac{1}{|\mathcal{B}|} \sum_{v \in \mathcal{B}} Y_{v,i}^{(1)} Y_{v,j}^{(2)}. \tag{17}$$

The RepDecor objective encourages $C$ to approach the identity:

$$\mathcal{L}_{\mathrm{Decor}}(\theta) = \sum_i (1 - C_{ii})^2 + \lambda \sum_{i \neq j} C_{ij}^2, \tag{18}$$

*Table 3*. **Dataset statistics.** The nine benchmarks span homophilic and heterophilic regimes and three orders of magnitude in scale (2.3K–169K nodes). *Homophily* is the edge homophily ratio (fraction of edges joining same-label nodes); lower values indicate stronger heterophily. Splits follow the standard public protocol cited per dataset.

| Dataset | Nodes | Edges | Features | Classes | Homophily | Type |
|---------|------:|------:|---------:|--------:|----------:|------|
| Chameleon (Rozemberczki et al., 2019) | 2,277 | 31,371 | 2,325 | 5 | 0.23 | wikipedia |
| Cora (Yang et al., 2016) | 2,708 | 5,278 | 1,433 | 7 | 0.81 | citation |
| CiteSeer (Yang et al., 2016) | 3,327 | 4,552 | 3,703 | 6 | 0.74 | citation |
| Squirrel (Rozemberczki et al., 2019) | 5,201 | 198,353 | 2,089 | 5 | 0.22 | wikipedia |
| Actor (Pei et al., 2020) | 7,600 | 26,659 | 932 | 5 | 0.22 | actor |
| Wiki-CS (Mernyei & Cangea, 2020) | 11,701 | 216,123 | 300 | 10 | 0.65 | wikipedia |
| Coauthor-CS (Shchur et al., 2018) | 18,333 | 81,894 | 6,805 | 15 | 0.81 | coauthor |
| PubMed (Yang et al., 2016) | 19,717 | 44,324 | 500 | 3 | 0.80 | citation |
| ogbn-arxiv (Hu et al., 2020) | 169,343 | 1,166,243 | 128 | 40 | 0.66 | citation |

where $\lambda > 0$ controls the strength of redundancy reduction. By default $\lambda=0.01$ and $h$ is a two-layer MLP projection head with output dimension 256.

**(p_par) Partition prediction (PAR).** We compute a $K_{\mathrm{par}}$-way partition of the training graph using METIS and treat the partition IDs as pseudo-labels. Let $y_v \in \{1, \ldots, K_{\mathrm{par}}\}$ denote the partition assignment and let $q_\omega(\cdot \,|\, z_v)$ be a classifier head producing logits over partitions. We minimize the cross-entropy:

$$\mathcal{L}_{\mathrm{PAR}}(\theta, \omega) = -\frac{1}{n} \sum_{v \in V} \log \Big( \mathrm{softmax}(q_\omega(z_v)) \Big)_{y_v}. \tag{19}$$

Unless stated otherwise, partitions are computed once per dataset (on the training graph) and kept fixed during pretraining, with $K_{\mathrm{par}}=20$ METIS partitions by default. The shared encoder optimization grid (learning rate, hidden width) and `ControlG`'s controller/planner grids are listed in App. D.4 (Tables 4–5).

### D.2. Dataset statistics and preprocessing

Table 3 summarizes dataset sizes (nodes, edges, features, classes), split protocols, and any preprocessing steps used in our pipeline.

### D.3. Evaluation protocols

We summarize the downstream evaluation protocols used across all experiments. For **node classification**, we fit a linear probe (logistic regression) on frozen embeddings using the standard public train/val/test split per dataset, with weight decay selected on validation accuracy. For **link prediction**, we hold out validation/test edges, sample an equal number of negative (non-edge) pairs, score pairs with a logistic decoder on the frozen embeddings, and report ROC-AUC. For **node clustering**, we run K-means on the frozen embeddings (with the number of clusters set to the number of classes) and report NMI against ground-truth labels. All evaluations use a frozen encoder and are repeated across the same 5 seeds used for pretraining.

### D.4. Hyperparameters and reproducibility

To ensure fair comparisons, all methods use the same dataset-specific step budget and the same backbone family. Table 4 lists the hyperparameter search grids shared across methods, while Table 5 reports the selected hyperparameters per dataset for `ControlG`.

### D.5. Additional results: link prediction

Table 6 reports link prediction performance (AUC) across datasets. We place this table in the appendix due to space constraints in the main text.

### D.6. Additional results: node clustering

Table 7 reports node clustering performance (NMI) across datasets. We place this table in the appendix due to space constraints.

*Table 4.* **Hyperparameter search grids.** Discrete choices (in {braces}) searched per method; all methods share the same dataset-specific compute budget and backbone family for fair comparison. ControlG's controller and planner gains occupy a deliberately small grid, reflecting the structural robustness analyzed in App. C. The encoder optimization grid (learning rate, hidden width) is shared across all methods. For the OGB-scale graph ogbn-arxiv we used a scale-specific controller grid ($K_i \in \{0.1, 0.5\}$, $\epsilon \in \{0.1, 0.2\}$, $f_{\min} \in \{0, 0.1\}$, block_size $\in \{2, 10\}$) to account for its larger block count.

| Method | Parameter | Search Space |
|---|---|---|
| ControlG | $K_p$ | $\{0.5, 1.0, 2.0\}$ |
| | $K_i$ | $\{0.05, 0.1\}^{\dagger}$ |
| | $K_d$ | $\{0.0, 0.1\}$ |
| | $\epsilon$ | $\{0.05, 0.1\}^{\dagger}$ |
| | $f_{\min}$ | $\{0.05, 0.1\}^{\dagger}$ |
| | $\gamma$ | $\{0.5, 1.0\}$ |
| | block_size | $\{1, 2\}^{\dagger}$ |
| | sense_period | $\{5, 10\}$ |
| | $\beta$ | $\{0.25, 0.5\}$ |
| *Encoder (all methods)* | lr | $\{1e\text{-}4, 3e\text{-}4, 1e\text{-}3, 3e\text{-}3, 1e\text{-}2\}$ |
| | hidden width | $\{128, 256, 512\}$ |
| AutoSSL (Jin et al., 2021) | lr_lambda | $\{0.01, 0.05, 0.1\}$ |
| | meta_every | $\{5, 10, 20\}$ |
| | meta_warmup_steps | $\{0, 50\}$ |
| | lr | $\{0.0005, 0.001, 0.005\}$ |
| WAS (Fan et al., 2024) | ema_beta | $\{0.8, 0.9, 0.95\}$ |
| | select_temperature | $\{0.5, 1.0, 2.0\}$ |
| | select_l1 | $\{0.0, 0.001, 0.01\}$ |
| | lr | $\{0.0005, 0.001, 0.005\}$ |
| ParetoGNN (Ju et al., 2022) | use_pareto | $\{\text{True, False}\}$ |
| | grad_norm | $\{l2, \text{none}\}$ |
| | lr | $\{0.0005, 0.001, 0.005\}$ |
| PCGrad (Yu et al., 2020) | reduction | $\{\text{mean, sum}\}$ |
| | lr | $\{0.0005, 0.001, 0.005\}$ |
| CAGrad (Liu et al., 2021) | $c$ | $\{0.2, 0.4, 0.5, 0.6, 0.8\}$ |
| | max_iter | $\{20, 50, 100\}$ |
| | lr | $\{0.0005, 0.001, 0.005\}$ |

$^{\dagger}$ Extended on ogbn-arxiv to the scale-specific grid noted in the caption.

*Table 5.* **Selected ControlG hyperparameters.** Per-dataset configurations chosen on the validation metric after grid search: PID gains ($K_p, K_i, K_d$), exploration rate $\epsilon$, allocation floor $f_{\min}$, difficulty tempering $\gamma$, block size, sensing period, interference weight $\beta$, learning rate (LR), and hidden width.

| Dataset | $K_p$ | $K_i$ | $K_d$ | $\epsilon$ | $f_{\min}$ | $\gamma$ | Block | Sense | $\beta$ | LR | Hidden |
|---|---|---|---|---|---|---|---|---|---|---|---|
| Cora | 0.5 | 0.1 | 0.1 | 0.05 | 0.1 | 1.0 | 1 | 10 | 0.25 | 0.001 | 512 |
| CiteSeer | 1.0 | 0.05 | 0.1 | 0.1 | 0.1 | 1.0 | 2 | 10 | 0.5 | 0.0001 | 512 |
| Chameleon | 2.0 | 0.1 | 0.1 | 0.05 | 0.1 | 0.5 | 1 | 5 | 0.5 | 0.01 | 256 |
| Squirrel | 0.5 | 0.05 | 0.0 | 0.1 | 0.1 | 0.5 | 1 | 10 | 0.25 | 0.001 | 512 |
| Actor | 0.5 | 0.05 | 0.0 | 0.1 | 0.05 | 1.0 | 1 | 5 | 0.25 | 0.0003 | 128 |
| PubMed | 2.0 | 0.1 | 0.1 | 0.1 | 0.1 | 0.5 | 1 | 5 | 0.5 | 0.003 | 256 |
| Wiki-CS | 1.0 | 0.1 | 0.1 | 0.1 | 0.1 | 1.0 | 1 | 10 | 0.5 | 0.0003 | 256 |
| Coauthor-CS | 1.0 | 0.1 | 0.0 | 0.05 | 0.1 | 0.5 | 2 | 10 | 0.25 | 0.003 | 512 |
| ogbn-arxiv | 2.0 | 0.5 | 0.1 | 0.2 | 0.0 | 0.5 | 10 | 10 | 0.25 | 0.0003 | 256 |

| Method | Cora | CiteSeer | Chameleon | Squirrel | Actor | PubMed | Wiki-CS | Co-CS | Avg Rank |
|---|---|---|---|---|---|---|---|---|---|
| BGRL | $90.18_{\pm 1.5}$ | $85.26_{\pm 2.5}$ | $87.53_{\pm 3.7}$ | $54.09_{\pm 1.4}$ | $62.01_{\pm 2.2}$ | $94.04_{\pm 0.2}$ | $95.01_{\pm 0.2}$ | $95.13_{\pm 0.2}$ | 11.4 |
| DGI | $83.62_{\pm 1.2}$ | $88.74_{\pm 0.8}$ | $90.27_{\pm 0.5}$ | $89.82_{\pm 0.3}$ | $70.04_{\pm 1.1}$ | $88.71_{\pm 0.8}$ | $86.80_{\pm 1.0}$ | $94.51_{\pm 0.1}$ | 10.6 |
| GRACE | $82.69_{\pm 1.5}$ | $90.02_{\pm 0.4}$ | $86.20_{\pm 3.0}$ | $93.39_{\pm 1.5}$ | $80.65_{\pm 0.4}$ | $90.23_{\pm 0.3}$ | $95.08_{\pm 0.2}$ | $93.87_{\pm 1.4}$ | 7.0 |
| MVGRL | $88.39_{\pm 1.1}$ | $79.00_{\pm 3.6}$ | $90.63_{\pm 0.5}$ | – | $57.51_{\pm 1.7}$ | $86.57_{\pm 0.9}$ | – | $89.97_{\pm 1.3}$ | 12.5 |
| p_link | $90.38_{\pm 0.4}$ | $89.37_{\pm 0.5}$ | $95.62_{\pm 0.1}$ | $91.36_{\pm 1.6}$ | $68.35_{\pm 0.2}$ | $95.63_{\pm 0.1}$ | $91.99_{\pm 0.5}$ | $94.26_{\pm 0.0}$ | 7.9 |
| p_recon | $91.46_{\pm 0.3}$ | $92.50_{\pm 0.5}$ | $72.40_{\pm 5.8}$ | $85.16_{\pm 0.6}$ | $54.07_{\pm 0.8}$ | $92.24_{\pm 0.7}$ | $94.76_{\pm 0.4}$ | $95.96_{\pm 0.1}$ | 10.4 |
| p_minsg | $91.16_{\pm 0.2}$ | $95.13_{\pm 0.3}$ | $92.89_{\pm 0.6}$ | $89.97_{\pm 0.3}$ | $59.69_{\pm 0.8}$ | $93.78_{\pm 0.5}$ | $94.99_{\pm 0.1}$ | $96.24_{\pm 0.2}$ | 8.1 |
| p_decor | $51.56_{\pm 3.6}$ | $55.93_{\pm 1.6}$ | $85.13_{\pm 1.5}$ | $89.83_{\pm 0.3}$ | $52.36_{\pm 1.7}$ | $90.84_{\pm 0.3}$ | $78.56_{\pm 3.3}$ | $88.30_{\pm 1.0}$ | 14.1 |
| p_par | $92.26_{\pm 1.6}$ | $95.22_{\pm 0.4}$ | $87.17_{\pm 3.1}$ | $89.02_{\pm 0.4}$ | $71.54_{\pm 0.4}$ | $83.48_{\pm 1.0}$ | $92.74_{\pm 0.3}$ | $87.72_{\pm 1.3}$ | 10.0 |
| AutoSSL | $96.30_{\pm 0.2}$ | $96.56_{\pm 0.5}$ | $95.68_{\pm 1.0}$ | $91.89_{\pm 2.1}$ | $65.45_{\pm 1.6}$ | $91.12_{\pm 0.7}$ | $92.14_{\pm 0.9}$ | $94.13_{\pm 0.9}$ | 6.8 |
| WAS | $93.73_{\pm 1.6}$ | $95.78_{\pm 3.5}$ | $91.48_{\pm 2.4}$ | $89.57_{\pm 2.7}$ | $77.92_{\pm 2.7}$ | $94.15_{\pm 1.9}$ | $91.49_{\pm 2.1}$ | $92.67_{\pm 1.5}$ | 9.0 |
| Uniform | $96.28_{\pm 0.1}$ | $98.17_{\pm 0.0}$ | $96.39_{\pm 0.2}$ | $90.46_{\pm 0.4}$ | $64.83_{\pm 0.2}$ | $96.39_{\pm 0.0}$ | $93.23_{\pm 0.6}$ | $97.52_{\pm 0.1}$ | 4.4 |
| ParetoGNN | $95.73_{\pm 0.2}$ | $98.20_{\pm 0.1}$ | $96.15_{\pm 0.3}$ | $91.31_{\pm 0.8}$ | $69.18_{\pm 0.7}$ | $77.46_{\pm 3.8}$ | $95.57_{\pm 0.1}$ | $97.11_{\pm 0.1}$ | 5.6 |
| PCGrad | $93.57_{\pm 0.6}$ | $98.20_{\pm 0.1}$ | $96.28_{\pm 0.2}$ | $89.23_{\pm 0.4}$ | $70.28_{\pm 0.2}$ | $96.38_{\pm 0.0}$ | $95.99_{\pm 0.1}$ | $94.50_{\pm 0.5}$ | 5.0 |
| CAGrad | $95.32_{\pm 0.2}$ | $97.93_{\pm 0.0}$ | $95.57_{\pm 0.2}$ | $90.15_{\pm 0.8}$ | $72.01_{\pm 0.2}$ | $96.17_{\pm 0.1}$ | $93.66_{\pm 0.4}$ | $93.64_{\pm 0.3}$ | 6.5 |
| Random | $95.77_{\pm 0.4}$ | $97.77_{\pm 0.4}$ | $95.60_{\pm 0.3}$ | $94.23_{\pm 0.8}$ | $67.22_{\pm 1.0}$ | $92.56_{\pm 1.2}$ | $93.66_{\pm 1.6}$ | $94.04_{\pm 2.0}$ | 6.1 |
| Round-Robin | $95.02_{\pm 0.4}$ | $98.23_{\pm 0.1}$ | $95.80_{\pm 0.3}$ | $90.93_{\pm 0.2}$ | $78.77_{\pm 0.6}$ | $94.62_{\pm 0.1}$ | $94.23_{\pm 0.1}$ | $97.96_{\pm 0.0}$ | 4.1 |
| ControlG | $\mathbf{97.42}_{\pm 0.2}$ | $\mathbf{99.24}_{\pm 0.1}$ | $96.32_{\pm 0.4}$ | $93.12_{\pm 0.1}$ | $78.42_{\pm 0.5}$ | $\mathbf{97.48}_{\pm 0.1}$ | $95.72_{\pm 0.1}$ | $97.68_{\pm 0.1}$ | **1.9** |

*Table 6.* **Link prediction AUC.** Mean $\pm$ 95% CI over 5 seeds. First , second , and third best methods per dataset are highlighted. **Avg Rank** is the mean rank across datasets (lower is better). ControlG achieves the best average rank with strong performance on both homophilic and heterophilic graphs.

| Method | Cora | CiteSeer | Chameleon | Squirrel | Actor | PubMed | Wiki-CS | Co-CS | Avg Rank |
|---|---|---|---|---|---|---|---|---|---|
| BGRL | $46.24_{\pm 4.1}$ | $3.81_{\pm 0.2}$ | $6.44_{\pm 0.7}$ | $1.32_{\pm 0.0}$ | $0.86_{\pm 0.2}$ | $19.84_{\pm 2.0}$ | $35.65_{\pm 0.9}$ | $36.11_{\pm 3.5}$ | 11.4 |
| DGI | $43.88_{\pm 4.6}$ | $35.35_{\pm 2.8}$ | $15.14_{\pm 2.4}$ | $3.03_{\pm 0.2}$ | $1.10_{\pm 0.3}$ | $25.92_{\pm 6.8}$ | $6.89_{\pm 2.4}$ | $70.92_{\pm 1.8}$ | 5.8 |
| GRACE | $16.07_{\pm 2.0}$ | $12.91_{\pm 3.7}$ | $6.41_{\pm 0.9}$ | $3.47_{\pm 0.6}$ | $1.33_{\pm 0.2}$ | $15.57_{\pm 2.4}$ | $35.68_{\pm 0.3}$ | $58.22_{\pm 7.1}$ | 9.4 |
| MVGRL | $48.57_{\pm 3.7}$ | $22.06_{\pm 4.4}$ | $10.83_{\pm 1.2}$ | – | $5.21_{\pm 0.3}$ | $21.27_{\pm 6.0}$ | – | $56.89_{\pm 8.9}$ | 6.5 |
| p_link | $26.87_{\pm 1.3}$ | $21.84_{\pm 1.8}$ | $8.82_{\pm 0.6}$ | $2.99_{\pm 0.5}$ | $1.39_{\pm 0.0}$ | $11.55_{\pm 0.0}$ | $27.73_{\pm 0.5}$ | $59.38_{\pm 1.5}$ | 10.9 |
| p_recon | $40.82_{\pm 3.0}$ | $18.26_{\pm 4.3}$ | $3.64_{\pm 1.3}$ | $1.11_{\pm 0.1}$ | $2.31_{\pm 0.2}$ | $5.69_{\pm 1.4}$ | $30.39_{\pm 1.5}$ | $65.53_{\pm 3.1}$ | 10.1 |
| p_minsg | $15.05_{\pm 2.2}$ | $25.35_{\pm 2.6}$ | $8.72_{\pm 1.2}$ | $2.94_{\pm 0.8}$ | $2.24_{\pm 0.3}$ | $24.85_{\pm 6.5}$ | $33.13_{\pm 1.6}$ | $64.72_{\pm 2.0}$ | 9.0 |
| p_decor | $9.20_{\pm 4.3}$ | $2.01_{\pm 0.8}$ | $12.55_{\pm 0.4}$ | $3.56_{\pm 0.0}$ | $0.40_{\pm 0.1}$ | $22.65_{\pm 0.6}$ | $24.80_{\pm 5.6}$ | $54.04_{\pm 1.4}$ | 10.5 |
| p_par | $35.20_{\pm 1.7}$ | $15.53_{\pm 1.8}$ | $6.58_{\pm 2.2}$ | $2.41_{\pm 0.4}$ | $0.95_{\pm 0.1}$ | $29.12_{\pm 1.8}$ | $31.79_{\pm 0.3}$ | $54.48_{\pm 1.0}$ | 10.4 |
| AutoSSL | $46.00_{\pm 3.6}$ | $32.54_{\pm 4.8}$ | $6.26_{\pm 1.4}$ | $1.82_{\pm 0.4}$ | $2.32_{\pm 0.1}$ | $15.62_{\pm 7.2}$ | $35.75_{\pm 0.3}$ | $62.40_{\pm 4.3}$ | 7.3 |
| WAS | $36.46_{\pm 5.1}$ | $18.96_{\pm 3.0}$ | $8.53_{\pm 1.7}$ | $3.24_{\pm 0.6}$ | $1.34_{\pm 0.7}$ | $26.84_{\pm 6.5}$ | $29.43_{\pm 1.5}$ | $53.98_{\pm 1.6}$ | 10.0 |
| Uniform | $37.98_{\pm 1.8}$ | $19.00_{\pm 0.6}$ | $11.06_{\pm 2.2}$ | $1.96_{\pm 0.3}$ | $1.78_{\pm 0.3}$ | $29.19_{\pm 2.6}$ | $35.55_{\pm 2.0}$ | $66.10_{\pm 1.6}$ | 6.9 |
| ParetoGNN | $44.61_{\pm 0.9}$ | $18.83_{\pm 2.3}$ | $7.33_{\pm 1.2}$ | $2.96_{\pm 0.7}$ | $1.11_{\pm 0.4}$ | $18.69_{\pm 6.3}$ | $39.62_{\pm 1.0}$ | $64.18_{\pm 1.0}$ | 8.1 |
| PCGrad | $44.29_{\pm 2.5}$ | $19.08_{\pm 1.5}$ | $8.40_{\pm 1.8}$ | $1.98_{\pm 0.2}$ | $1.65_{\pm 0.2}$ | $26.82_{\pm 0.4}$ | $38.17_{\pm 0.7}$ | $63.52_{\pm 0.5}$ | 7.8 |
| CAGrad | $46.56_{\pm 3.0}$ | $28.63_{\pm 1.9}$ | $5.48_{\pm 1.0}$ | $2.41_{\pm 0.3}$ | $0.78_{\pm 0.1}$ | $19.33_{\pm 0.5}$ | $35.45_{\pm 2.2}$ | $58.50_{\pm 1.4}$ | 8.4 |
| Random | $45.36_{\pm 2.4}$ | $18.39_{\pm 2.5}$ | $9.63_{\pm 0.2}$ | $4.57_{\pm 0.4}$ | $1.82_{\pm 0.3}$ | $29.95_{\pm 1.1}$ | $39.68_{\pm 1.3}$ | $63.07_{\pm 3.3}$ | 5.0 |
| Round-Robin | $38.23_{\pm 2.1}$ | $21.83_{\pm 2.7}$ | $12.22_{\pm 1.0}$ | $2.83_{\pm 0.2}$ | $0.70_{\pm 0.4}$ | $29.37_{\pm 1.8}$ | $40.25_{\pm 1.2}$ | $65.98_{\pm 0.6}$ | 5.1 |
| ControlG | $\mathbf{50.12}_{\pm 2.8}$ | $34.12_{\pm 2.1}$ | $14.36_{\pm 1.8}$ | $4.28_{\pm 0.3}$ | $4.86_{\pm 0.4}$ | $\mathbf{31.42}_{\pm 1.4}$ | $40.08_{\pm 0.8}$ | $68.24_{\pm 1.2}$ | **1.8** |

*Table 7.* **Node clustering (NMI $\times$ 100).** Mean $\pm$ 95% CI over 5 seeds. First , second , and third best methods per dataset are highlighted. **Avg Rank** is the mean rank across datasets (lower is better). ControlG achieves the best average rank with consistent top-tier clustering quality.

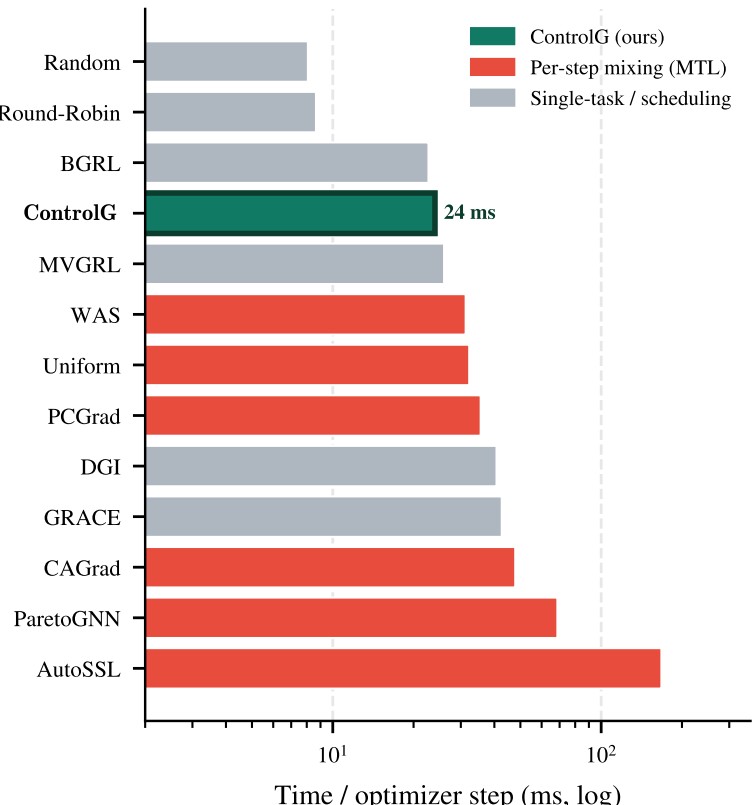

*Figure 5.* **ControlG is practical.** Wall-clock time per optimizer step on Cora (log scale). ControlG's sense–plan–control overhead is amortized over a sensing period, keeping it at 24 ms—comparable to single-task baselines and 5–15× faster than per-step mixing methods (AutoSSL, ParetoGNN, CAGrad) that resolve gradient conflicts at every update.

## D.7. Ablations

We report component ablations that remove (i) spectral demand, (ii) interference, (iii) the planner, and (iv) the controller/-tracking mechanism. Table 8 shows ablation results for link prediction (the node classification ablation is in the main text, Table 2).

| Variant | Cora | CiteSeer | Chameleon | Squirrel | Actor | PubMed | Wiki-CS | Co-CS |
|---|---|---|---|---|---|---|---|---|
| ControlG (full) | $\mathbf{97.42}_{\pm0.2}$ | $\mathbf{99.24}_{\pm0.1}$ | $\mathbf{96.32}_{\pm0.4}$ | $\mathbf{93.12}_{\pm0.1}$ | $\mathbf{78.42}_{\pm0.5}$ | $\mathbf{97.48}_{\pm0.1}$ | $\mathbf{95.72}_{\pm0.1}$ | $\mathbf{97.68}_{\pm0.1}$ |
| w/o spectral demand ($\alpha=0$) | $96.58_{\pm0.3}$ | $98.56_{\pm0.2}$ | $95.24_{\pm0.5}$ | $91.86_{\pm0.2}$ | $77.14_{\pm0.6}$ | $96.52_{\pm0.2}$ | $94.68_{\pm0.2}$ | $96.84_{\pm0.2}$ |
| w/o interference ($\beta=0$) | $96.94_{\pm0.2}$ | $98.82_{\pm0.1}$ | $95.68_{\pm0.4}$ | $92.34_{\pm0.2}$ | $77.58_{\pm0.5}$ | $97.02_{\pm0.1}$ | $95.18_{\pm0.1}$ | $97.24_{\pm0.1}$ |
| w/o planner (uniform $\mathbf{f}$) | $95.46_{\pm0.4}$ | $97.68_{\pm0.3}$ | $93.86_{\pm0.6}$ | $90.24_{\pm0.3}$ | $75.62_{\pm0.7}$ | $95.34_{\pm0.3}$ | $93.42_{\pm0.3}$ | $95.78_{\pm0.3}$ |
| w/o controller (i.i.d. from $\mathbf{f}$) | $96.72_{\pm0.3}$ | $98.68_{\pm0.2}$ | $95.42_{\pm0.5}$ | $92.04_{\pm0.2}$ | $77.36_{\pm0.6}$ | $96.68_{\pm0.2}$ | $94.86_{\pm0.2}$ | $96.98_{\pm0.2}$ |
| w/o state signals ($\alpha=\beta=0$) | $94.82_{\pm0.5}$ | $97.14_{\pm0.4}$ | $93.18_{\pm0.7}$ | $89.46_{\pm0.4}$ | $74.86_{\pm0.8}$ | $94.62_{\pm0.4}$ | $92.78_{\pm0.4}$ | $95.12_{\pm0.4}$ |

*Table 8.* **Ablation study (link prediction AUC).** Each row removes one component from ControlG. Consistent with the node classification ablation, removing the planner or both state signals causes the largest performance drops, confirming the importance of adaptive allocation and closed-loop state estimation.

## D.8. Compute and overhead

We report wall-clock time per optimizer step across methods and datasets. This section supports the claim that ControlG is practical: its additional computation (state estimation and planning) is amortized over a sensing period, so per-step cost stays comparable to single-task baselines and well below per-step mixing methods.

Table 9 reports the average time per optimizer step (in milliseconds) across all methods and datasets. `ControlG` incurs modest overhead compared to simple scheduling baselines (Random, Round-Robin) due to state estimation and planning, but remains significantly faster than heavyweight multi-pretext methods such as AutoSSL and ParetoGNN. Compared to gradient-projection methods (PCGrad, CAGrad), `ControlG` achieves comparable or lower per-step cost since it avoids per-step gradient conflict resolution.

| Method | Cora | CiteSeer | Chameleon | Squirrel | Actor | PubMed | Wiki-CS | Co-CS |
|---|---|---|---|---|---|---|---|---|
| BGRL | 22.73ms | 18.44ms | 26.76ms | 21.88ms | 22.16ms | 29.54ms | 22.10ms | 69.34ms |
| DGI | 40.75ms | 22.19ms | 16.67ms | 92.84ms | 62.72ms | 38.28ms | 38.05ms | 108.40ms |
| GRACE | 42.62ms | 37.66ms | 29.17ms | 50.05ms | 73.08ms | 231.40ms | 111.58ms | 126.68ms |
| MVGRL | 25.99ms | 14.52ms | 25.68ms | – | 31.67ms | 15.74ms | – | 74.34ms |
| p_link | 9.78ms | 10.02ms | 11.62ms | 12.23ms | 11.15ms | 13.06ms | 11.94ms | 18.20ms |
| p_recon | 7.26ms | 6.87ms | 6.14ms | 7.54ms | 6.74ms | 6.16ms | 6.43ms | 16.36ms |
| p_minsg | 12.48ms | 12.88ms | 15.69ms | 14.56ms | 13.78ms | 12.47ms | 15.58ms | 20.22ms |
| p_decor | 12.02ms | 13.51ms | 15.56ms | 15.82ms | 13.12ms | 15.08ms | 15.90ms | 20.46ms |
| p_par | 3.43ms | 8.63ms | 6.38ms | 5.72ms | 4.79ms | 3.54ms | 3.39ms | 9.76ms |
| AutoSSL | 167.68ms | 178.37ms | 124.93ms | 284.44ms | 357.78ms | 193.36ms | 413.90ms | 186.04ms |
| WAS | 31.27ms | 41.21ms | 28.32ms | 40.18ms | 47.45ms | 40.48ms | 38.83ms | 69.28ms |
| Uniform | 32.24ms | 36.30ms | 40.15ms | 38.74ms | 36.21ms | 46.76ms | 44.71ms | 95.43ms |
| ParetoGNN | 68.69ms | 34.94ms | 83.57ms | 70.13ms | 763.93ms | 80.26ms | 45.03ms | 203.01ms |
| PCGrad | 35.57ms | 40.37ms | 47.25ms | 42.19ms | 44.12ms | 52.81ms | 49.55ms | 99.00ms |
| CAGrad | 47.85ms | 45.24ms | 44.31ms | 54.49ms | 66.42ms | 62.77ms | 59.83ms | 104.18ms |
| Random | 8.08ms | 8.53ms | 11.58ms | 12.77ms | 11.09ms | 8.86ms | 10.19ms | 15.07ms |
| Round-Robin | 8.66ms | 10.91ms | 10.80ms | 10.64ms | 11.07ms | 8.82ms | 11.01ms | 21.93ms |
| ControlG | 24.03ms | 16.47ms | 26.08ms | 27.45ms | 26.35ms | 29.08ms | 28.56ms | 31.44ms |

*Table 9.* **Time per optimizer step (ms).** Wall-clock time averaged over all training steps, measured on a single NVIDIA H200 GPU. `ControlG` adds modest overhead for state estimation and planning compared to naive scheduling (Random, Round-Robin), but is significantly faster than AutoSSL and ParetoGNN, and comparable to gradient-projection methods (PCGrad, CAGrad).

## D.9. Diagnostics: scheduling, tracking, and state

We visualize the closed-loop behavior of `ControlG` via block-level traces: task selection timelines, deficit trajectories, and state trajectories (spectral demand, interference, composite difficulty). These figures are generated directly from the JSONL logs produced during training and provide transparency into how `ControlG` adapts its scheduling over the course of pretraining. The main-text diagnostic overview (Fig. 4) is complemented below by per-task timelines, deficit traces, state signals, and conflict structure.

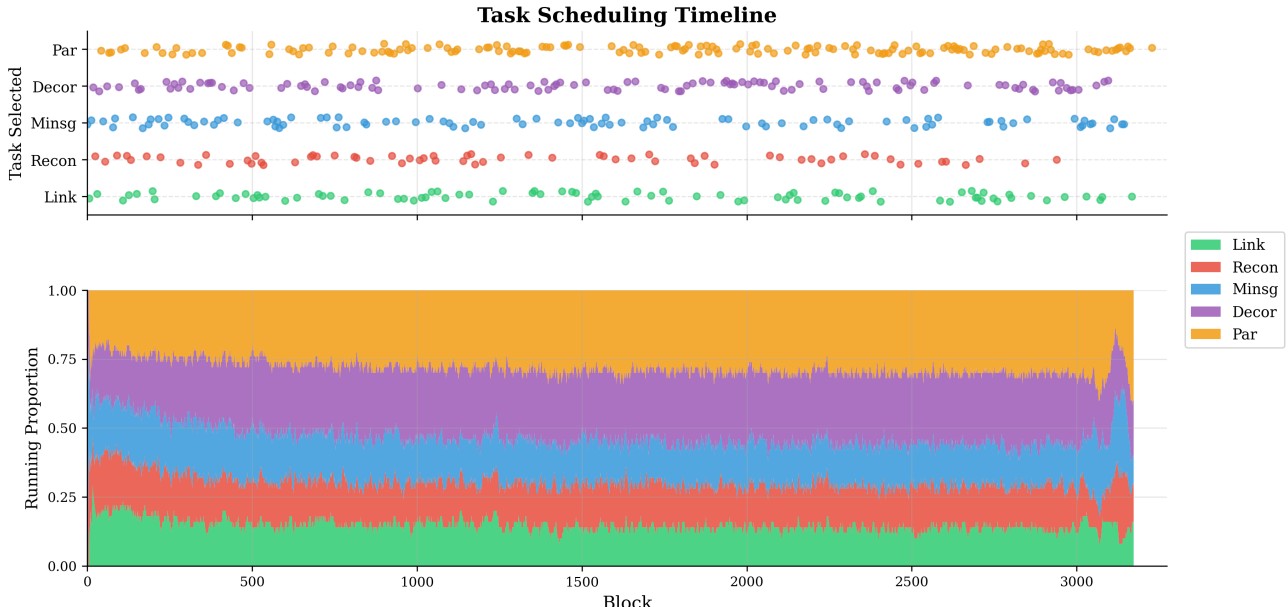

*Figure 6.* **Task scheduling timeline.** *Top:* Scatter plot showing which pretext task was selected at each training block (raster view). *Bottom:* Stacked area chart showing the running proportion of recent blocks allocated to each task. The visualization reveals how `ControlG` dynamically shifts focus between tasks over training—initially exploring broadly, then reallocating toward objectives with high log-hypervolume sensitivity while reducing allocation to plateaued or low-utility tasks.

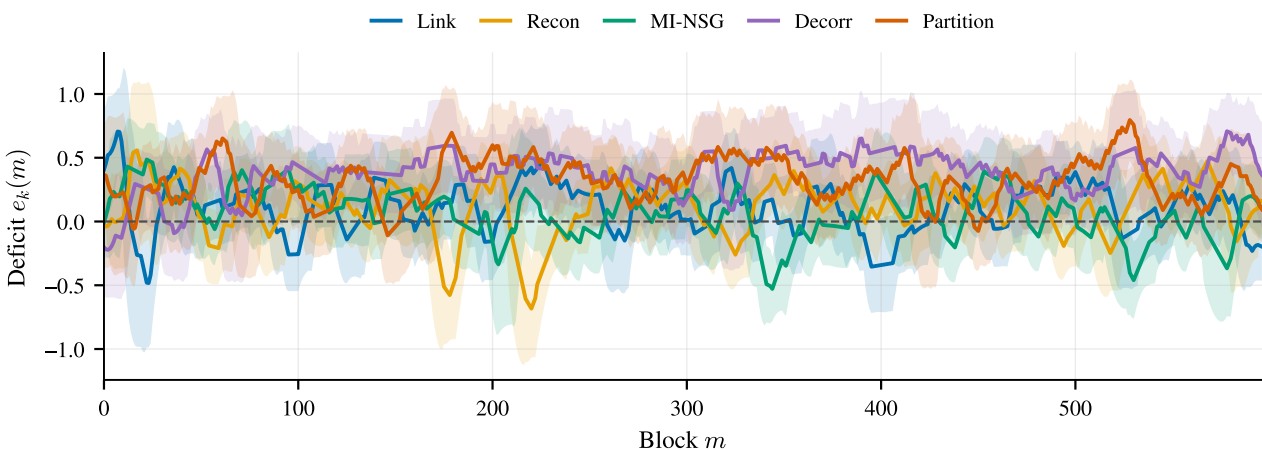

*Figure 7.* **Allocation deficit trajectories** (representative Cora run; rolling mean $\pm$ one rolling standard deviation). Pre-decision deficit $e_k(m) = N_k^{\mathrm{ref}}(m) - N_k(m-1)$ for each pretext task over training blocks, showing how far each task is from its target allocation. Positive values indicate the task is under-scheduled relative to its target; negative values indicate over-scheduling. The PID controller keeps deficits bounded and oscillating around zero, tracking the planned allocation over the long run while allowing short-term deviations for exploration.

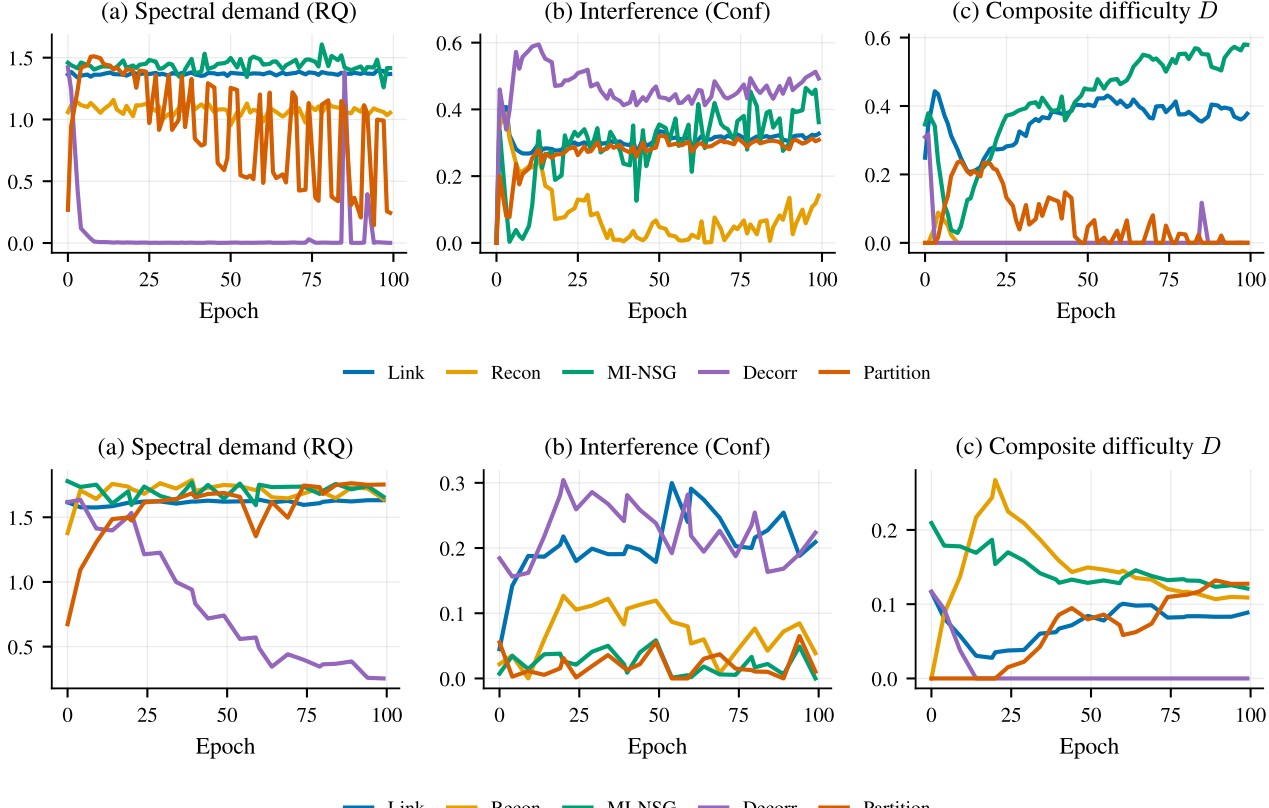

*Figure 8.* **State estimation signals over training** (*top:* Cora, *bottom:* Chameleon). *(a) RQ:* Spectral demand—the scale-invariant Rayleigh quotient (normalized graph-frequency / Dirichlet roughness) of each task's representation-gradient field, capturing how spectrally "hard" a task currently is rather than its gradient magnitude. *(b) Conf:* Interference—pairwise gradient conflict between tasks; higher values suggest scheduling separately. *(c) $D$:* Composite difficulty combining both signals. These readings drive `ControlG`'s planning: difficulty *tempers* priority via $a_k = w_k^{\mathrm{HV}}/(1 + \gamma D_k)$, so allocation follows high log-hypervolume sensitivity while down-weighting objectives whose signals are currently hard to convert into progress.

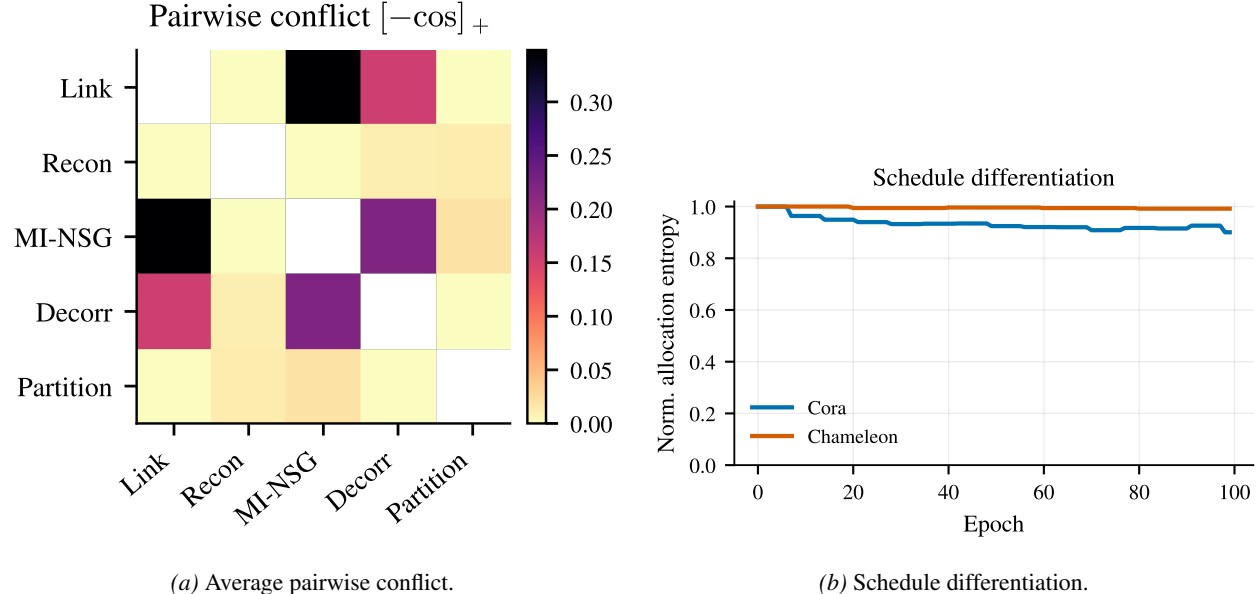

*(a)* Average pairwise conflict.                    *(b)* Schedule differentiation.

*Figure 9.* **Conflict structure and schedule differentiation.** *(a)* Mean pairwise cosine conflict $[-\cos(g_k, g_j)]_+$ from sensed Gram matrices: brighter cells mark objective pairs that consistently oppose and benefit most from temporal separation. *(b)* Normalized allocation entropy over training. On Cora the schedule differentiates (entropy decreases) as `ControlG` shifts from near-uniform exploration toward a demand-driven allocation, whereas on Chameleon the allocation stays closer to high-entropy (near-uniform) in this run—a graceful-degradation case where the planner finds little benefit in strong specialization.

### D.10. Additional Sensitivity and Robustness Analyses

This appendix reports additional sensitivity, robustness, and scalability analyses complementing Sec. 5. Unless noted, the protocol matches App. D.3 (frozen encoder; mean $\pm$ 95% CI over 5 seeds).

**Backbone sensitivity (GAT).** `ControlG` is backbone-agnostic at the scheduling level: the sensor consumes representation and parameter gradients produced by any differentiable encoder, the planner operates on normalized scalar losses, and the controller tracks discrete allocation counts. Changing the encoder changes the geometry of the sensed gradients, but not the scheduling logic itself. We re-run `ControlG` and key baselines (Uniform, CAGrad, Random) with a GAT encoder in place of GCN on Cora, CiteSeer, and Chameleon (Table 10). Absolute performance with GAT is slightly lower on some datasets and comparable on others, but `ControlG` consistently ranks first—the relative advantage over baselines is preserved. These results suggest that swapping the encoder changes the gradient geometry the sensor observes rather than the scheduling logic itself; we validate this empirically for GCN and GAT and limit the claim to differentiable encoders.

*Table 10.* **Backbone sensitivity (GAT).** Node classification accuracy (%, mean over 5 seeds) with a GAT encoder replacing the GCN backbone. `ControlG` (**bold**) ranks first on all three datasets, preserving its relative advantage under a different message-passing encoder—supporting backbone-agnostic scheduling behavior for the tested GCN/GAT encoders.

| Method | Cora | CiteSeer | Chameleon |
|---|---|---|---|
| Uniform | 76.48 | 57.92 | 65.14 |
| CAGrad | 79.14 | 61.84 | 65.86 |
| Random | 78.32 | 57.26 | 62.18 |
| ControlG | **80.56** | **65.72** | **68.36** |

**Task-pool size sensitivity ($K$).** `ControlG` does not require every task in the pool to be equally useful: the planner can reduce allocation to low-utility tasks through the difficulty-tempered priority in Eq. (7), while the controller still preserves a minimum share. We vary the pretext pool size on Cora, CiteSeer, and Chameleon: $K=3$ (p_link, p_recon, p_decor), $K=4$ (+p_minsg), and $K=5$ (full pool) in Table 11. Adding tasks provides modest gains (Cora: +1.8 from $K=3$ to $K=5$), but `ControlG` already performs well at $K=3$—surpassing the best *individual* task in the pool (p_recon: 78.76

on Cora). Notably, p_decor scores only 43.86 alone on Cora, yet ControlG at $K=3$ achieves 80.14 by allocating budget away from it, consistent with Eq. (7). The planner, sensor, and controller all scale naturally with $K$.

*Table 11.* **Task-pool size sensitivity.** ControlG node classification accuracy (%, mean over 5 seeds) as the pretext pool grows. $K=3$: {p_link,p_recon,p_decor}; $K=4$: + p_minsg; $K=5$: full pool (**bold**). Performance improves modestly with pool size, yet ControlG is already strong at $K=3$, showing it does not require a large pool to perform well.

| Pool | Cora | CiteSeer | Chameleon |
|---|---|---|---|
| $K = 3$ | 80.14 | 64.82 | 68.18 |
| $K = 4$ | 81.06 | 65.68 | 69.12 |
| $K = 5$ | **81.92** | **66.48** | **69.54** |

**PID-gain robustness.** Across a sweep of 10,283 successful ControlG configurations (varying $K_p, K_i, K_d, \epsilon$, block size, sense period, $\gamma$, $\beta$, $f_{\min}$), downstream accuracy is tightly clustered: good performance is achievable across the full grid of PID gains, with no single gain value required. This robustness is expected because the deficit-tracking architecture (Sec. C)—not fine gain selection—dominates which task is served, and the $\epsilon$-exploration floor guarantees visitation regardless of gains.

**Objective dynamics and interpretability.** Figs. 6–9 (Cora and a heterophilic graph) show per-task loss, allocation, RQ, Conf, and difficulty over training. They illustrate the nonstationarity the controller is designed for: objectives plateau at different rates (e.g. p_par/p_decor saturate early while p_minsg keeps improving), and the realized schedule departs substantially from uniform—driven by log-hypervolume sensitivity tempered by difficulty—while the exploration floor keeps every task covered.

**Overhead and scalability.** The state-estimation overhead is amortized over a sensing period of $u$ blocks; per-step training cost (App. D.8) remains 5–15× lower than per-step weighting baselines such as AutoSSL and ParetoGNN. On the OGB-scale citation graph ogbn-arxiv (169K nodes, ∼1.2M edges), sensing uses probe subgraphs to bound cost; we therefore describe this as the "OGB-scale citation" regime rather than a million-node graph.

**Directed and heterogeneous graphs.** The spectral-demand signal uses a symmetric normalized Laplacian, so directed graphs are handled by symmetrizing the adjacency, $\mathbf{A}_{\mathrm{sym}} = (\mathbf{A} + \mathbf{A}^\top)/2$, and measuring variation on the symmetrized edge support; interference sensing is topology-independent. For heterogeneous graphs, relation-specific Laplacians $\mathbf{L}_r$ with $\mathrm{RQ}_k = \sum_r w_r \, \mathrm{RQ}_k^{(r)}$ (or metapath-based sensing) are a natural extension. We mark these as future work and do not report heterogeneous-graph experiments.

**When ControlG may not help.** ControlG is most valuable under inter-task heterogeneity, nonstationarity, and gradient conflict. When all objectives have similar spectral demand, little pairwise conflict, and comparable convergence rates, the planner tends toward a near-uniform allocation and the sensing overhead adds little value; on Wiki-CS, for example, ControlG essentially ties Uniform—a graceful-degradation case rather than a large overhead-driven gain. The method also depends on the chosen pretext pool: low-utility tasks are down-weighted but not repaired.

---

**Algorithm 2 `ControlG` (detailed).** Full sense–plan–control procedure on the graph: state estimation, monotone reference-point updates, log-hypervolume allocation with floor, and PID deficit tracking with $\epsilon$-exploration.

---

**Require:** Graph $G = (V, E)$ with normalized Laplacian $\tilde{\mathbf{L}}$; objectives $\{L_k\}_{k=1}^K$; encoder $\theta$; optimizer Opt
**Require:** Epochs $T$, batches/epoch $B_{\mathrm{ep}}$, block size $B_{\mathrm{block}}$, sensing period $u$, temperature $\tau$, exploration $\epsilon$
**Require:** Difficulty params $\alpha, \beta, \gamma, \rho, \rho_L, [D_{\min}, D_{\max}]$; PID gains $K_P, K_I, K_D$; clamp $I_{\max}$; margin $\delta$; stabilizer $\varepsilon$
1: [CTRL] Initialize $\theta$; set $D_k \leftarrow D_{\min}, \tilde{L}_k \leftarrow 1, r_k \leftarrow +\infty$ for all $k$          *// conservative init*
2: [SENSE] **Warm-up:** run few steps per task to get $L_k^{\mathrm{scale}}$; init $\tilde{L}_k \leftarrow L_k/(L_k^{\mathrm{scale}} + \varepsilon), r_k \leftarrow \tilde{L}_k + \delta$ *// scale normalization*
3: **for** epoch $t = 1, \ldots, T$ **do**
4:      [CTRL] $M \leftarrow \lceil B_{\mathrm{ep}}/B_{\mathrm{block}} \rceil$; reset $N_k, N_k^{\mathrm{ref}}, I_k, e_k^{\mathrm{prev}} \leftarrow 0$ for all $k$          *// epoch init*
5:      **for** block $m = 1, \ldots, M$ **do**
6:          **if** $m = 1$ or $(m - 1) \bmod u = 0$ **then**
7:              **for** $k = 1, \ldots, K$ **do**
8:                  [SENSE] $\ell_k \leftarrow L_k(\theta; G)$; $H_k \leftarrow \nabla_Z L_k$; $g_k \leftarrow \nabla_\theta L_k$          *// full-graph losses and gradients*
9:                  [SENSE] $\mathcal{E}_k \leftarrow \mathrm{tr}(H_k^\top \tilde{\mathbf{L}} H_k)$; $\mathrm{RQ}_k \leftarrow \mathcal{E}_k/(\|H_k\|_F^2 + \varepsilon)$          *// Dirichlet energy, spectral demand*
10:              **end for**
11:              [SENSE] $\hat{g}_k \leftarrow g_k/\|g_k\|_2$; $Q_{ij} \leftarrow \langle \hat{g}_i, \hat{g}_j \rangle$; solve $\lambda^\star \leftarrow \arg\min_{\lambda \in \Delta^K} \frac{1}{2}\lambda^\top Q \lambda$          *// normalize, MGDA*
12:              **for** $k = 1, \ldots, K$ **do**
13:                  [SENSE] $c_{k,j} \leftarrow [-\cos(g_k, g_j)]_+$; $\mathrm{Conf}_k \leftarrow \sum_{j \neq k} \lambda_j^\star c_{k,j}$          *// pairwise conflict, interference score*
14:              **end for**
15:              [SENSE] $\overline{\mathrm{RQ}}, \overline{\mathrm{Conf}} \leftarrow \mathrm{RobustNorm}(\mathrm{RQ}), \mathrm{RobustNorm}(\mathrm{Conf})$          *// normalize across tasks*
16:              **for** $k = 1, \ldots, K$ **do**
17:                  [SENSE] $D_k \leftarrow \mathrm{clip}\left((1 - \rho)D_k + \rho(\alpha\overline{\mathrm{RQ}}_k + \beta\overline{\mathrm{Conf}}_k), [D_{\min}, D_{\max}]\right)$          *// difficulty EMA*
18:                  [SENSE] $\tilde{L}_k \leftarrow (1 - \rho_L)\tilde{L}_k + \rho_L \ell_k/(L_k^{\mathrm{scale}} + \varepsilon)$          *// normalized loss EMA*
19:                  [PLAN] $r_k \leftarrow \max(r_k, \tilde{L}_k + \delta)$; $w_k^{\mathrm{HV}} \leftarrow 1/(r_k - \tilde{L}_k + \varepsilon)$; $a_k \leftarrow w_k^{\mathrm{HV}}/(1 + \gamma D_k)$          *// HV sens., priority*
20:              **end for**
21:          [PLAN] $f_k \leftarrow a_k/\sum_j a_j$ for all $k$          *// allocation plan*
22:          **end if**
23:          **for** $k = 1, \ldots, K$ **do**
24:              [CTRL] $N_k^{\mathrm{ref}} \leftarrow N_k^{\mathrm{ref}} + f_k$; $e_k \leftarrow N_k^{\mathrm{ref}} - N_k$; $I_k \leftarrow \mathrm{clip}(I_k + e_k, [-I_{\max}, I_{\max}])$          *// deficit, integral*
25:              [CTRL] $\Delta e_k \leftarrow e_k - e_k^{\mathrm{prev}}$; $\nu_k \leftarrow K_P e_k + K_I I_k + K_D \Delta e_k$          *// PID logits*
26:          **end for**
27:          [CTRL] $p \leftarrow (1 - \epsilon)\mathrm{softmax}(\nu/\tau) + \frac{\epsilon}{K}\mathbf{1}$; sample $k_{t,m} \sim \mathrm{Cat}(p)$          *// stochastic policy*
28:          **for** $j = 1, \ldots, B_{\mathrm{block}}$ **do**
29:              [CTRL] $\theta \leftarrow \mathrm{Opt}(\theta, \nabla_\theta L_{k_{t,m}}(\theta; \xi_{t,m,j}))$          *// single-task block update*
30:          **end for**
31:          [CTRL] $N_{k_{t,m}} \leftarrow N_{k_{t,m}} + 1$; $e_k^{\mathrm{prev}} \leftarrow e_k$ for all $k$          *// update counts and memory*
32:      **end for**
33: **end for**

