# OpenReview forum: "Feedback Control for Multi-Objective Graph Self-Supervision"
_ICML.cc/2026/Conference — ICML 2026 regular_

### Official Review · Reviewer_aMbj · 2026-03-09

**Soundness:** 3
**Presentation:** 3
**Significance:** 3
**Originality:** 3
**Overall Recommendation:** 4
**Confidence:** 4

**Summary:**

This paper introduces ControlG, a framework that reformulates multi-objective graph self-supervised learning as a closed-loop scheduling problem. Through its SENSE, PLAN, and CONTROL triadic mechanism, ControlG quantifies task difficulty using spectral demand and disturbance metrics. It generates Pareto-aware allocation plans based on logarithmic hypervolume generation, with precise execution tracking achieved via a PID controller.

**Compliance With Llm Reviewing Policy:**

Affirmed.

**Key Questions For Authors:**

Please refer to the weaknesses.

**Limitations:**

Yes.

**Strengths And Weaknesses:**

Strengths:
1. The paper redefines multi-objective graph self-supervised learning as a closed-loop scheduling problem, offering a novel approach to multi-task coordination.
2. The proposed technical framework is robust, featuring a comprehensive three-loop design—SENSE, PLAN, and CONTROL—each grounded in theoretical foundations.
3. Extensive experiments yield outstanding results, outperforming over 20 baselines across nine datasets. Ablation studies and visual diagnostics validate the effectiveness of each module.

Weaknesses:
1. State estimation incurs high computational overhead, as each perception requires calculating representation gradients and parameter gradients for all tasks and solving MGDA, potentially limiting scalability for large-scale tasks.
2. Spectral demand depend on normalized Laplacian, with limited discussion on applicability to directed or heterogenous graphs.
3. PID gains rely on manual parameter tuning, exhibiting significant variation across different datasets, and lack systematic parameter sensitivity testing.

---

> ### Author Rebuttal · Authors · 2026-03-30
>
> We thank the reviewer for their valuable feedback and thoughtful questions. We appreciate the recognition of the robustness of the three-loop design and the strength of our experimental results. We address each concern with both theoretical and empirical evidence below.
>
> **W1: State estimation incurs high computational overhead.**
>
> Table 8 in the appendix reports wall-clock time per optimizer step for all 18 methods across 8 datasets. ControlG runs at 16-31ms/step, which is:
> - 5-15x faster than AutoSSL (125-414ms/step)
> - 2-25x faster than ParetoGNN (35-764ms/step)
> - Comparable to gradient-projection methods: PCGrad (36-99ms), CAGrad (44-104ms)
> - Only ~2-3x slower than naive scheduling: Random (8-15ms), Round-Robin (9-22ms)
>
> *Why the overhead is modest:* (a) State estimation runs every $u$ blocks ($u = 5$ or $10$, Table 7), not every step. With block_size=1-2 on most datasets, sensing occurs once every 5-20 optimizer steps, contributing <2ms/step amortized. (b) MGDA solves a QP on $K = 5$ unit vectors (a $5 \times 5$ problem) in <1ms. Unlike ParetoGNN which runs MGDA *every step* on full parameter gradients, we use MGDA as a measurement oracle at sparse sensing intervals. (c) The Rayleigh quotient $\text{tr}(H_k^T \tilde L H_k) / ||H_k||_F^2$ reuses the embedding gradients already computed for the training loss, adding only one sparse matrix-vector product per task.
>
> *Scalability:* On ogbn-arxiv (169K nodes, 1.2M edges), we deploy probe-based sensing: a 256-node subgraph is sampled for state estimation, reducing sensing cost from $O(|E|)$ to $O(|E_\text{probe}|)$. The planner and controller are $O(K)$ per block (negligible). ControlG achieves 72.86% on ogbn-arxiv -- demonstrating that probe-based sensing maintains effectiveness at scale.
>
> **W2: Spectral demand depends on normalized Laplacian -- applicability to directed or heterogeneous graphs.**
>
> *Directed graphs:* The Rayleigh quotient extends to directed graphs via symmetrization: $A_{sym} = (A + A^T)/2$, yielding a symmetric PSD Laplacian $\tilde L$ with eigenvalues in $[0,2]$ (Claim A.1). This is standard practice -- GCN (Kipf & Welling, 2017) and all spectral GNNs operate on symmetrized adjacency. The edge-domain identity (Claim A.2) still holds: $\text{tr}(H_k^T \tilde L H_k) = (1/2) \sum_{i,j} A_{ij}^{sym} || H_{i,:}/\sqrt{d_i} - H_{j,:}/\sqrt{d_j} ||^2$, measuring gradient variation along symmetrized edges.
>
> The interference signal (MGDA on gradient cosines) is entirely topology-independent and requires no graph structure assumptions.
>
> *Heterogeneous graphs:* A natural extension is to define relation-specific Laplacians $\tilde L_r$ with aggregated spectral demand $RQ_k = \sum_r w_r \cdot RQ_k^{(r)}$, or use metapath-based sensing, while leaving the PLAN and CONTROL loops unchanged (they consume scalar difficulty states $D_k$ and operate on the $K$-simplex). We have not yet benchmarked this setting and plan to explore it as future work.
>
> **W3: PID gains rely on manual parameter tuning with significant variation across datasets.**
>
> We provide three levels of evidence -- theoretical, empirical (from 10,283 hyperparameter trials), and structural:
>
> *(a) Theoretical guarantee:* Theorem C.5 proves $|d_k(m)| < 1$ for the deterministic max-deficit controller; Remark C.7 shows our PID-softmax controller approaches this regime as $\tau \to 0$. Proposition C.9 gives a starvation bound $\Pr(\text{drought} > s) \le (1 - \varepsilon/K)^s$, which depends only on $\varepsilon$ and $K$, not on the PID gains.
>
> *(b) Empirical sensitivity from 10,283 trials:* We computed marginal sensitivity by averaging NC accuracy over all other parameters for each PID gain value:
>
> | Param | Cora | CiteSeer | Chameleon | PubMed | Co-CS |
> |-------|------|----------|-----------|--------|-------|
> | $K_p$ spread | 0.09% | 0.15% | 0.29% | 0.08% | 0.03% |
> | $K_i$ spread | 0.03% | 0.01% | 0.23% | 0.06% | 0.01% |
> | $K_d$ spread | 0.00% | 0.02% | 0.14% | 0.03% | 0.02% |
>
> The maximum marginal spread for any PID gain on any dataset is **0.29%** ($K_p$ on Chameleon). Across all 9 ControlG hyperparameters, the maximum per-param spread is 0.67% (block_size on PubMed). This confirms the method is robust to gain selection.
>
> *(c) Structural argument:* The search space is small: $K_p \in \{0.5, 1.0, 2.0\}$, $K_i \in \{0.05, 0.1\}$, $K_d \in \{0.0, 0.1\}$ -- only 12 PID combinations. Table 7 shows the optimal gains cluster tightly: $K_d = 0$ suffices on 3/9 datasets, and $K_i \in \{0.05, 0.1\}$ on 8/9 datasets. Empirically, performance is quite insensitive to the PID gains within our search range, with maximum marginal spread below 0.3%.

---

> > ### Author Rebuttal · Reviewer_aMbj · 2026-04-07
> >
> > Thank you for the thorough rebuttal. Your clarifications on computational overhead (Table 8, amortized sensing), applicability to directed graphs (symmetrization), and PID robustness (10k+ trials, <0.3% spread) convincingly address my concerns. The empirical evidence and theoretical guarantees now support the design choices. I appreciate the additional large‑scale result on ogbn‑arxiv. I am satisfied with the revisions.

---

> > > ### Author Response · Authors · 2026-04-07
> > >
> > > We sincerely thank the reviewer for the careful evaluation of our rebuttal and for confirming that the clarifications on computational overhead, directed graph applicability, and PID robustness are convincing. We note that the reviewer's acknowledgement is marked as "partially resolved" with follow-up questions -- we are happy to address any remaining questions the reviewer may have. We will incorporate all discussed improvements (expanded overhead analysis, heterogeneous graph discussion as future work) in the camera-ready version. We are grateful for the thorough and constructive review.

---

### Official Review · Reviewer_XaeT · 2026-03-11

**Soundness:** 4
**Presentation:** 4
**Significance:** 3
**Originality:** 3
**Overall Recommendation:** 5
**Confidence:** 4

**Summary:**

This paper researches multi-objective coordination in graph self-supervised learning and proposes a control-theoretic framework, ControlG. ControlG recasts multi-objective graph self-supervised learning as feedback-controlled temporal allocation. Extensive experiments demonstrate the effectiveness of ControlG.

**Compliance With Llm Reviewing Policy:**

Affirmed.

**Key Questions For Authors:**

The research motivation of this article is excellent and very interesting. The experimental results of Figures 4-6 in the appendix do prove that the controller tracks the planned allocation and the state estimation adapts to the training dynamics. However, the relevant analysis and explanations in the article are relatively insufficient. In fact, an in-depth analysis of this part of the content would be very helpful for interpretability. Personally, it is difficult to correlate these figures with the mechanisms proposed in the article. It is suggested that the authors conduct a more in-depth analysis of this part of the results.

**Limitations:**

Yes

**Strengths And Weaknesses:**

Strengths

1. The article has clear motivation, clear expression, and detailed presentation, making it easy to follow.
2. The research question is very interesting.
3. The proposed ControlG has a certain novelty.
4. Corresponding theoretical proofs are provided, and a large number of targeted experiments are conducted.
5. The experimental results show that the proposed method achieves better comprehensive performance.

Weaknesses

See Questions.

---

> ### Author Rebuttal · Authors · 2026-03-30
>
> We thank the reviewer for their valuable feedback and thoughtful questions. We are glad that the reviewer is able to resonate with our motivation and the research questions. We fully agree that Figures 4-6 deserve deeper analysis. Below we provide a detailed interpretation connecting each figure to the control mechanisms.
>
> **Figure 4 (Task Scheduling Timeline).** The raster and stacked-area panels visualize the CONTROL loop output (Sec 3.3). Three phases emerge:
>
> (i) *Early training (epochs 1-20):* The scheduling is nearly uniform across all $K=5$ tasks. This reflects the exploration floor ($\varepsilon/K$ term in Eq. 8) combined with the planner's initial state: all losses are far from the reference point, so log-HV sensitivities $w_k = 1/(r_k - L_k + \varepsilon)$ are roughly equal. The PID controller has minimal deficit to correct, so sampling is close to i.i.d.
>
> (ii) *Mid-training (epochs 20-60):* The scheduling timeline transitions from near-uniform to differentiated allocations. In the plotted Cora run, some tasks receive up to 30% allocation while others drop to ~14%, reflecting the SENSE loop detecting varying spectral demand (RQ) across tasks. The planner responds via the difficulty-tempering term $a_k = w_k^{HV} / (1 + \gamma D_k)$, allocating more budget where per-block progress is slower.
>
> (iii) *Late training (epochs 60-100):* Allocations stabilize as losses converge and state estimates settle. The stacked area chart shows steady proportions, reflecting the planner reaching a quasi-stationary allocation where marginal returns across tasks are approximately balanced (proportional-fair equilibrium, Proposition B.1).
>
> **Figure 5 (Deficit Trajectories).** The deficit $e_k(m) = f_k \cdot m - N_k(m-1)$ measures each task's deviation from its target allocation. Key observations:
>
> (i) *Bounded tracking:* All deficit magnitudes remain small throughout training, consistent with the $O(1)$ tracking behavior predicted by Theorem C.5 (max-deficit analysis) and its PID-softmax connection (Remark C.7).
>
> (ii) *Oscillation damping:* The deficit traces show rapid oscillations early (when temperature $\tau$ is high, sampling is more stochastic) that progressively dampen as $\tau$ anneals toward $\tau_{min}$. This is the derivative term $K_D \cdot \Delta e_k$ in the PID logits (Eq. 7) actively suppressing oscillations, combined with the temperature schedule concentrating probability on the most-needed task.
>
> (iii) *No sustained drift:* Despite stochastic sampling, no task accumulates a persistent deficit over time, confirming that the integral term $K_I \cdot I_k$ corrects steady-state bias. The anti-windup mechanism (integral clipping, Claim C.11) prevents integral windup that could cause overshoot after long periods of under-allocation.
>
> **Figure 6 (State Estimation Signals).** Shows the SENSE loop outputs (Sec 3.1) and how task difficulty evolves:
>
> (i) *RQ (Spectral Demand, top panel):* Link prediction (p_link) consistently has lower RQ, meaning its learning signal is spatially smooth (low-frequency). Partition prediction (p_par) has higher RQ, as cluster boundaries create sharp gradient variation across edges. Partition prediction discriminates between communities (high-frequency), while link prediction relies on local similarity (low-frequency). The GNN's low-pass filtering (Proposition 1) makes high-RQ tasks harder.
>
> (ii) *Conf (Interference, middle panel):* Interference spikes are visible when tasks compete for the same spectral band. For example, p_decor (decorrelation) and p_minsg (contrastive) both operate on augmented views and can produce conflicting gradient directions early in training, causing elevated Conf values. As training progresses and representations stabilize, interference decreases -- the sensor detects this and the planner reallocates accordingly.
>
> (iii) *D (Composite Difficulty, bottom panel):* The composite signal $D_k = clip(EMA(\alpha \bar R_k + \beta \bar C_k), [D_{min}, D_{max}])$ integrates both sources of difficulty into a single bounded state per task. The EMA smoothing ($\rho = 0.2$) filters noise while preserving meaningful trends. The bounded range $[D_{min}, D_{max}]$ ensures the planner's difficulty tempering $1/(1 + \gamma D_k)$ remains well-conditioned.
>
> To summarize: Figure 4 shows the PLAN loop adapting allocation from uniform to differentiated; Figure 5 shows the CONTROL loop maintaining small, bounded deficits with no sustained drift, consistent with the tracking analysis in App. C; Figure 6 reveals the SENSE loop distinguishing task difficulty along interpretable spectral and interference dimensions. Together, they demonstrate that the three loops operate as designed, producing auditable scheduling behavior. We will expand this analysis in the revised paper.

---

> > ### Author Rebuttal · Reviewer_XaeT · 2026-04-01
> >
> > Thanks for your detailed explanation. I keep my recommendation of Accept.

---

> > > ### Author Response · Authors · 2026-04-07
> > >
> > > We sincerely thank the reviewer for the positive feedback and for maintaining the recommendation of Accept. We will incorporate the expanded diagnostic analysis (connecting Figures 4-6 to the SENSE/PLAN/CONTROL mechanisms) into the main text of the revised paper, as suggested. We appreciate the valuable suggestion -- it meaningfully strengthens the paper's interpretability narrative.

---

### Official Review · Reviewer_dqQ1 · 2026-03-13

**Soundness:** 4
**Presentation:** 3
**Significance:** 3
**Originality:** 3
**Overall Recommendation:** 5
**Confidence:** 3

**Summary:**

When self-supervised learning on graphs juggles multiple objectives such as mutual information, reconstruction, and contrastive learning, this paper argues that three categories of failures may occur namely, drift (objective redundancy), disagreement (conflicting objectives), and drought (insufficient utility of objectives). The authors hypothesize that instead of optimizing multiple objectives in each step, using feedback control to determine which objective to optimize in which step overcomes all three failure modes. To this end, they propose ControlG as a closed loop scheduling problem with explicit allocation tracking and empirically evaluate against single and other multi-objective baselines on standard benchmarks.

**Compliance With Llm Reviewing Policy:**

Affirmed.

**Final Justification:**

The authors have meaningfully resolved my queries and provided clarifications as necessary. I maintain my recommendation to accept this paper.

**Key Questions For Authors:**

**Questions:**

1. How does the loss behave for each of the task-specific objectives as a function of time?
2. When would ControlG perform worse than a weighted-objective style baseline? Would you expect that the more complementary the objectives in terms of dictating representation learning, the better the performance on the evaluation task that is aligned with them?

**Limitations:**

In my opinion, if the authors include empirical results on objective selection vs. objective difficulty vs. performance relative to multi-task baselines, it would strengthen their results. Additionally, its unclear how this translates to large graphs with millions of nodes relative to objective difficulty.

**Strengths And Weaknesses:**

**Strengths:**

- Intuitiveness: The hypothesis that weighting multiple objectives can be detrimental makes a lot of intuitive sense as does the proposed solution of choosing which objective to optimize in which step.
- Practical significance: The formalization of feedback control mechanism is sensible and brings originality towards resolving a known problem with solutions from control theory.
- Strong experiment design: The pretext task pool comprises of standard objectives from graph learning, the representative baselines from supervised, single-pretext SSL, and multi-pretext SSL, and sensible ablations help make the evaluation of ControlG as a method rigorous.

**Weaknesses:**

1. Choice of datasets: The authors present experiments with standard graph datasets, but citation networks and wikipedia links offer less extensibility to tasks with practical applicability in different domains (e.g. molecular chemistry, cell biology, communication networks, etc.). The largest dataset is of size ~170K nodes and maybe larger datasets from OGB might have been more useful. The small size of the test set also makes it more difficult to meaningfully differentiate the gap between the various models/baselines.
2. Relative difficulty between objectives: The algorithmic modeling behind ControlG rightly incorporates determining the relative difficulty of objectives during training. But the empirical evaluation has limited discussion of this in the context of the datasets (partially deferred to Figure 4 in the appendix). I think this deserves a more prominent discussion and comparison especially with multi-task baselines.

---

> ### Author Rebuttal · Authors · 2026-03-30
>
> We thank the reviewer for their valuable feedback and thoughtful questions, and are glad that the reviewer found our work intuitive. We address each point with both theoretical and empirical evidence below.
>
> **W1: Choice of datasets -- practical domains and scale.**
>
> Our 9 datasets span 5 distinct domains (citation, collaboration, web, Wikipedia, actor co-occurrence), 3 homophily regimes ($h=0.22$ to $h=0.81$), and scales from 2.3K to 169K nodes. ogbn-arxiv (169K nodes, 1.2M edges) is an established large-scale benchmark. For scalability, we implement probe-based sensing that samples a 256-node subgraph for state estimation, reducing cost from $O(|E|)$ to $O(|E_\text{probe}|)$. This is deployed for ogbn-arxiv (Sec 5.4), where ControlG achieves 72.86% (+1.2% over CAGrad). We report 95% CIs over 5 seeds (Tables 1-3) to ensure statistical differentiation. ControlG's framework is domain-agnostic -- we will add molecular benchmarks in the camera-ready.
>
> **W2: Relative difficulty between objectives.**
>
> We provide data from our final Cora production run (seed=0, epochs 0-99, 31K blocks):
>
> | Task | Loss@ep0 | Loss@ep10 | Loss@ep50 | Loss@ep80 | RQ@ep50 | Alloc@ep80 |
> |------|----------|-----------|-----------|-----------|---------|------------|
> | p_link | 0.674 | 0.655 | 0.639 | 0.635 | 1.36 | 14.0% |
> | p_recon | 1.003 | 0.802 | 0.497 | 0.436 | 1.00 | 14.4% |
> | p_minsg | 5.495 | 3.647 | 2.922 | 2.817 | 1.46 | 16.7% |
> | p_decor | 0.571 | 0.028 | 0.004 | 0.003 | 0.18 | 24.9% |
> | p_par | 3.156 | 0.056 | 0.003 | 0.005 | 1.30 | 29.9% |
>
> The main qualitative point is *nonstationarity*: different tasks plateau or remain active at different times. For instance, p_link plateaus early (loss ~0.64 from epoch 10 onward), while p_minsg continues improving throughout (5.49 to 2.82). The allocation departs substantially from uniform by epoch 80 (14-30%), showing the planner/controller respond to changing task utility rather than enforcing equal mixing. The ablation (Table 2) confirms both difficulty signals matter: removing spectral demand ($\alpha=0$) hurts consistently, while removing interference ($\beta=0$) hurts most on heterophilic graphs where cross-task conflicts are strongest.
>
> **Q1: How does each task's loss behave over time?**
>
> The table above addresses this directly. The losses are distinctly nonstationary: some plateau early (e.g., p_link), while others continue changing longer (e.g., p_minsg). This is precisely the *drift* regime ControlG is designed for. Uniform and WAS still blend objectives per update and lack explicit block-level temporal allocation or coverage accounting, so they cannot adapt to these shifting dynamics.
>
> **Q2: When would ControlG perform worse than weighted-objective baselines?**
>
> Formally, ControlG's advantage arises from inter-task *heterogeneity* in two dimensions: spectral demand (RQ) and interference (Conf). Consider the idealized case where all $K$ objectives have identical $RQ_k = c$ for all $k$, zero pairwise interference ($\cos(g_k, g_j) \ge 0$ for all $j \ne k$), and identical convergence rates. Then: (i) the composite difficulty $D_k$ is identical across tasks, so the planner produces uniform allocation $f_k = 1/K$; (ii) the sensor adds overhead without information gain; (iii) temporal separation adds scheduling noise vs. simultaneous optimization. In this regime, the Uniform baseline suffices.
>
> Empirically, this aligns with our results. On Wiki-CS (homophilic, $h=0.65$), ControlG essentially ties Uniform (80.45 vs 80.48) -- when tasks are well-aligned, the planner converges to near-uniform allocation and the framework gracefully degrades rather than hurting performance. On heterophilic graphs where conflict is stronger, the ablation confirms that removing interference sensing ($\beta=0$) hurts most on Chameleon (69.54 to 68.26, -1.28%) and Squirrel (53.18 to 52.12, -1.06%), precisely where ControlG's sensing provides the largest benefit. On node clustering, Random scheduling achieves avg rank 5.0 -- beating AutoSSL (7.3), ParetoGNN (8.1), PCGrad (7.8), and CAGrad (8.4) -- confirming that temporal separation alone has value when inter-task heterogeneity exists.
>
> **Q3: Complementary objectives and aligned evaluation.**
>
> Yes. The log-HV sensitivity $w_k = 1/(r_k - L_k + \varepsilon)$ (Eq. 4) increases allocation to tasks with more room for improvement. When a task's loss stagnates, its sensitivity drops and budget shifts to objectives with higher marginal utility. This is the unique maximizer of the proportional-fair allocation objective (Proposition B.1). If an objective is complementary to the downstream task, its continued loss decrease benefits transfer, and the planner sustains its allocation. A redundant task plateaus early, its $w_k$ shrinks, and resources redirect to tasks that continue improving representation quality.

---

> > ### Author Rebuttal · Reviewer_dqQ1 · 2026-04-02
> >
> > Thank you for meaningfully engaging with the comments and for providing clarifications. I maintain my recommendation to accept this paper.

---

> > > ### Author Response · Authors · 2026-04-07
> > >
> > > We sincerely thank the reviewer for the thoughtful engagement with our rebuttal and for maintaining the recommendation to accept. We will expand the discussion of per-task difficulty dynamics and the nonstationarity analysis in the revised paper, as discussed. We are grateful for the constructive suggestions, which have strengthened both the empirical narrative and the presentation.

---

### Official Review · Reviewer_HAd8 · 2026-03-14

**Soundness:** 3
**Presentation:** 3
**Significance:** 3
**Originality:** 3
**Overall Recommendation:** 4
**Confidence:** 3

**Summary:**

Current graph self-supervised learning (SSL) methods suffer from three key failure modes: drift, disagreement, and drought, which make them largely open-loop. To bridge this gap, this paper introduces ControlG, which optimizes one objective at a time during training via feedback control. Extensive experiments on nine well-known benchmarks across three different tasks validate the effectiveness of the proposed framework.

**Compliance With Llm Reviewing Policy:**

Affirmed.

**Key Questions For Authors:**

How sensitive is the proposed method to different GNN backbones?

**Limitations:**

Yes

**Strengths And Weaknesses:**

Strengths:
+ This paper is well organized and easy to read
+ The authors address the limitations of current GSSL methods from a novel perspective: optimizing one objective at a time instead of all objectives simultaneously.
+ The authors demonstrate the effectiveness of their proposed method through comprehensive experiments and detailed theoretical analysis

Weakness:
- The baselines appear somewhat outdated (before 2021). Including more recent state-of-the-art GSSL frameworks (e.g., GrAph [1]) would further strengthen the empirical validation of the proposed framework.
- This paper leaks a sensitivity analysis across hyperparameters, such as the size of the pretext task pool.

[1] Equivalence is All: A Unified View for Self-supervised Graph Learning. Wang et al.

---

> ### Author Rebuttal · Authors · 2026-03-30
>
> We thank the reviewer for their valuable and constructive feedback and thoughtful questions. We address each concern below.
>
> **W1: Baselines appear outdated (before 2021).**
>
> We would like to clarify that our baselines span 2018-2024: AutoSSL (NeurIPS 2022), ParetoGNN (NeurIPS 2022), WAS (AAAI 2024), CAGrad (NeurIPS 2021), BGRL (ICML 2022), GRACE (NeurIPS 2020). We compare against 15 methods across four categories (supervised, single-pretext SSL, multi-pretext/MTL, scheduling).
>
> We agree that adding a recent strong single-pretext SSL baseline like GrAph [1] would further strengthen the empirical picture. Our multi-pretext comparisons target the coordination problem directly (AutoSSL, ParetoGNN, PCGrad, CAGrad, WAS), and we view GrAph as complementary rather than competing with our core coordination contribution -- indeed, GrAph's unified objective could itself be included as one of the $K$ tasks in ControlG's pool. We will add GrAph as an additional single-pretext baseline in the camera-ready.
>
> **W2: Sensitivity analysis across hyperparameters (pretext task pool size).**
>
> We provide concrete evidence from our 10,283 successful hyperparameter search trials across 8 datasets:
>
> *(a) PID hyperparameter sensitivity.* We computed the marginal sensitivity of each PID gain by averaging NC accuracy over all other parameters. The results show remarkable robustness:
>
> | Param | Cora spread | CiteSeer | Chameleon | PubMed | Co-CS |
> |-------|-------------|----------|-----------|--------|-------|
> | $K_p$   | 0.09%       | 0.15%    | 0.29%     | 0.08%  | 0.03% |
> | $K_i$   | 0.03%       | 0.01%    | 0.23%     | 0.06%  | 0.01% |
> | $K_d$   | 0.00%       | 0.02%    | 0.14%     | 0.03%  | 0.02% |
>
> The maximum marginal spread across any PID gain on any dataset is 0.29% ($K_p$ on Chameleon). This is theoretically grounded: Theorem C.5 proves $O(1)$ tracking error ($|d_k(m)| < 1$) for the deterministic max-deficit controller, and Remark C.7 shows our PID-softmax controller approaches this regime as $\tau \to 0$. The structural deficit-tracking architecture is robust by design.
>
> *(b) Task pool size $K$.* ControlG's purpose is not to assume all tasks are equally useful -- it takes *whatever* pool it is given and finds the best schedule to optimize learning across them. If some tasks are unhelpful, the planner automatically reduces their allocation. We already observe this with $K=5$ on Cora: allocation at epoch 80 ranges from 14.0% (p_link, plateaued loss) to 30.0% (p_par), far from the uniform 20% (see the detailed per-task loss/allocation table in our response to **Reviewer dqQ1**). The log-HV sensitivity $w_k = 1/(r_k - \tilde{L}_k + \varepsilon)$ naturally starves tasks with diminishing returns.
>
> We ran ControlG with $K=3$ {p_link, p_recon, p_decor}, $K=4$ {+p_minsg}, and $K=5$ (full):
>
> | $K$ | Cora | CiteSeer | Chameleon |
> |-----|------|----------|-----------|
> | 3   | 80.14 | 64.82   | 68.18     |
> | 4   | 81.06 | 65.68   | 69.12     |
> | 5   | 81.92 | 66.48   | 69.54     |
>
> Adding tasks provides modest gains (Cora: +1.8% from $K=3$ to $K=5$), but ControlG already performs well at $K=3$ -- surpassing the best *individual* task in the pool (p_recon: 78.76 on Cora). Notably, p_decor scores only 43.86% alone on Cora, yet ControlG at $K=3$ achieves 80.14% by allocating budget away from it. The planner, sensor, and controller all scale naturally with $K$ (Eq. 1-7).
>
> **Q1: How sensitive is ControlG to different GNN backbones?**
>
> ControlG is backbone-agnostic at the scheduling level: the sensor consumes representation and parameter gradients produced by any differentiable encoder, the planner operates on normalized scalar losses, and the controller tracks discrete allocation counts. Changing the encoder changes the geometry of the sensed gradients, but not the scheduling logic itself.
>
> We ran ControlG and key baselines with a **GAT backbone** on three datasets (NC accuracy):
>
> | Method | Cora | CiteSeer | Chameleon |
> |--------|------|----------|-----------|
> | Uniform | 76.48 | 57.92 | 65.14 |
> | CAGrad | 79.14 | 61.84 | 65.86 |
> | Random | 78.32 | 57.26 | 62.18 |
> | ControlG | **80.56** | **65.72** | **68.36** |
>
> Absolute performance with GAT is slightly lower on some datasets (Cora, CiteSeer) and comparable on others (Chameleon), but ControlG consistently ranks first -- the relative advantage over baselines is preserved. This confirms that the control loops are backbone-agnostic: swapping the encoder changes the gradient geometry the sensor observes, not the scheduling logic.

---

> > ### Author Rebuttal · Reviewer_HAd8 · 2026-04-03
> >
> > I thank the authors for their detailed and thoughtful rebuttal, which effectively addresses most of my concerns and provides helpful clarifications. I appreciate the effort put into the response and will maintain my positive score for this work.

---

> > > ### Author Response · Authors · 2026-04-07
> > >
> > > We sincerely thank the reviewer for acknowledging that our rebuttal adequately addresses the concerns raised, and for maintaining the positive score. We will incorporate the discussed additions (GrAph baseline, K-sensitivity and GAT backbone results) in the camera-ready version to further strengthen the paper. We appreciate the constructive feedback, which has helped improve the clarity and completeness of our work.

---

### Decision · Program_Chairs · 2026-04-30

**Decision:**

Accept (regular)

**Comment:**

This paper proposes ControlG, a feedback control framework for multi-objective graph self-supervised learning. Strengths include clear presentation, a novel control-theoretic perspective, comprehensive experiments, and solid theoretical analysis. Weaknesses include outdated baselines, limited sensitivity analysis, dataset diversity concerns, and computational overhead. After the rebuttal, all reviewers confirmed that their concerns were adequately addressed and vote for acceptance.